# Personalized Visual Representation Alignment for Generative Multimodal Recommendation

## Abstract

With the development of Vision-Language Models (VLMs) for multimodal understanding, recommender systems have increasingly leveraged them to process heterogeneous sources of user-interacted items for recommendation. By fine-tuning VLMs on user interaction data, prior works have adapted these models to capture user preferences, enabling personalized multimodal recommendation. Despite these advances, however, we identify two key limitations: 1) *the visual features directly extracted by vision encoders (e.g., CLIP) are insufficient for capturing personalized user preferences*, as such encoders are generally trained for generic visual perception rather than capturing user-specific preferences; and 2) *existing VLM-based methods often underutilize visual features of user-interacted items in later LLM layers*, relying instead on textual descriptions for recommendation—an unexpected bias that diminishes the contribution of visual features. To address these two limitations, we propose PerVRA, a VLM-based recommendation model consisting of a Personalized Visual Representation Learning (PVRL) module and a Personalized Multimodal Alignment (PMA) module. Specifically, we employ dual contrastive learning, where each module is equipped with its own contrastive objective: The PVRL module learns personalized visual features from user interaction history, while the PMA module enhances the contribution of visual features to the VLMs by explicitly aligning them with text features. Extensive experiments on real-world Amazon and H&M Fashion datasets demonstrate that PerVRA consistently outperforms strong VLM-based methods over diverse personalized tasks. Moreover, our ablation studies show that addressing these two limitations is critical for building effective VLM-based recommender systems.

## 1 Introduction

The rise of personalization has become a cornerstone in delivering user experiences that adapt to individual needs and preferences (Meng et al., 2020; Li et al., 2023; Geng et al., 2022; Deldjoo et al., 2022). In digital environments, users interact with heterogeneous data sources—such as reviews, product descriptions, prices, and images. As users increasingly rely on information from multiple modalities, it becomes essential to design personalization systems that unify these heterogeneous inputs and translate them into more accurate and meaningful recommendations. Advances in Vision-Language Models (Alayrac et al., 2022; Liu et al., 2023; Achiam et al., 2023; Anthropic, 2024) (VLMs) have created new opportunities across industries, introducing a paradigm shift in recommender systems. By leveraging heterogeneous multimodal data (*e.g.*, images and text descriptions) with VLMs, these systems can more effectively infer user preferences and support a variety of personalized recommendation tasks, including sequential recommendation, multimodal search, and personalized item selection (Geng et al., 2023; Wei et al., 2024a).

Specifically, several early works (Geng et al., 2023; Wei et al., 2024a) leverage pre-trained VLMs trained on large-scale web data for personalized recommendation. We identify two key limitations of naively applying VLMs to recommendations. First, in these methods, frozen vision encoders are used to extract visual features from item images, which are then provided as inputs to Large Language Models (LLMs). However, vision encoders pre-trained on web-scale image-text data (*e.g.*, CLIP Radford et al. (2021)) provide strong discriminative features for perception and classification

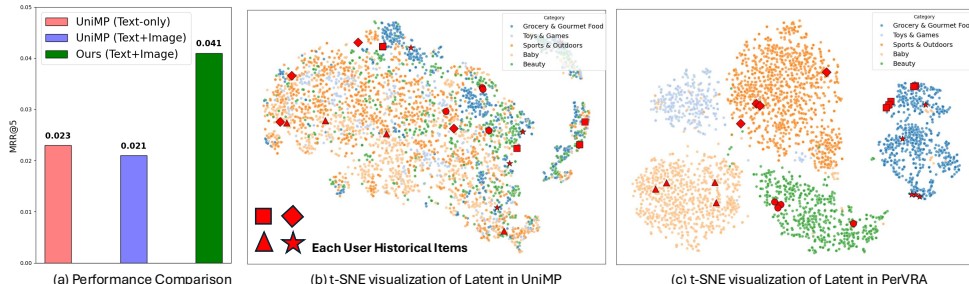

(a) Performance Comparison     (b) t-SNE visualization of Latent in UniMP     (c) t-SNE visualization of Latent in PerVRA

Figure 1: (a) Performance on Amazon Reviews. UniMP (Text-only) outperforms its multimodal variant, while PerVRA achieves the best results by fully leveraging visual information. (b, c) t-SNE visualization of item latents in VLMs. User historical items concentrate within relevant categories (e.g., beauty). PerVRA yields distinct clusters by both category and user preference.

but fall short in modeling individual user preferences, such as favored styles, colors, categories, or shapes. For example, if a user prefers kitchen-related items, objects like knives and frying pans should be embedded closer together, rather than treated as distinct and unrelated classes. Second, we observe that VLMs tend to overemphasize textual tokens while underutilizing visual information from user history, which is crucial for accurately capturing user preferences. This is consistent with prior work (Kaduri et al., 2025; Yoon et al., 2025), which shows that rich and detailed visual cues are often diminished in VLMs. In Fig. 1-(a), we compare previous VLM recommenders evaluated on text-only and multimodal data. While the text-only model slightly outperforms its multimodal variant, our proposed model with personalized visual representation alignment achieves a significant performance gain.

To address these two limitations, we propose a Personalized Visual Representation Alignment (PerVRA) framework for a VLM-based recommendation model to enhance multimodal recommendation. PerVRA consists of two objectives: (1) Personalized Visual Representation Learning (PVRL), which refines vision features by leveraging items from users' historical interactions, enabling the model to capture user-specific preferences such as styles, colors, or categories; and (2) Personalized Multimodal Alignment (PMA), which guides textual features in the LLM with the enhanced visual representations, jointly aligning vision and text in a personalized embedding space. Through these objectives, PerVRA effectively reshapes the multimodal space toward recommendation objectives, allowing personalized visual information to be fully utilized in the final prediction.

These objectives are achieved through our proposed dual personalized contrastive learning for recommendation, which is designed to jointly capture user-level preferences and recommendation-level distinctions. (1) User-oriented contrastive learning encourages all items from a user's historical interactions to be embedded close to each other in the latent space by treating them as positives. This design ensures that the learned visual representations reflect consistent user-specific preferences, such as favored styles, colors, or categories. (2) Recommendation-oriented contrastive learning complements this by contrasting the target recommendation items against the historical items, thereby learning to emphasize the distinguishing factors that make certain items more relevant for recommendation. Together, these two complementary objectives enforce both coherence within a user's historical interactions and discriminability for recommendation targets, resulting in a more personalized and effective multimodal alignment. In Fig. 1-(b), we compare latent item representations in VLMs between the baselines and our method. While the baseline exhibits entangled category representations with scattered user history items, our approach yields well-separated clusters by both categories and user histories.

We build PerVRA on top of the state-of-the-art multimodal recommendation model, UniMP (Wei et al., 2024a), and evaluate its effectiveness across a variety of personalized recommendation tasks. Our method is applied only during training and introduces no additional cost at inference time, while significantly enhancing recommendation performance. Overall, our key contributions can be summarized as follows:

- We analyze limitations of existing VLM-based recommender systems, showing that frozen vision encoders overlook user-specific preferences and that visual cues are often underutilized compared to textual tokens.

- We propose PerVRA, a Personalized Visual Representation Alignment framework, which introduces personalized visual representation learning (PVRL) and personalized multi-modal alignment (PMA) to refine vision features from user histories and align them with textual features in LLMs.

- We design a dual personalized contrastive learning scheme to jointly capture user-level coherence and recommendation-level discriminability, and demonstrate that integrating Per-VRA into UniMP yields consistent improvements across diverse recommendation tasks without additional inference cost, while generalizing to related tasks such as preference prediction, personalized image selection, and multimodal search.

## 2 RELATED WORK

**Multimodal Recommendation.** Research on multimodal recommendation has primarily focused on integrating various auxiliary item information, such as images and text, into sequential recommendation to enhance performance. Recently, sequential multimodal recommendation, which incorporates auxiliary information such as images and text, has gained increasing attention as a means to capture hidden user preferences within each modality, going beyond what can be achieved with collaborative filtering. An early work, VBPR (He & McAuley, 2016), incorporated visual features of items into latent factor models to improve personalized ranking performance. Subsequently, Pan et al. (2022) incorporates visual and textual information with multimodal meta-learning to improve recommendation accuracy. Similarly, MMMLP (Liang et al., 2023) designed an MLP-based architecture to directly incorporate multimodal sequences into sequential recommendation, enabling efficient and effective multimodal fusion. More recently, MP4SR (Zhang et al., 2024) combined multimodal pretraining with contrastive learning to capture correlations both between sequences and between sequences and items, demonstrating robust recommendation performance even under data-sparse environments. However, since these studies rely on each individually trained modality, a gap arises between modalities, which in turn negatively impacts recommendation performance.

**Personalized Recommendation using Vision-Language Models.** With the success of Vision-Language Models (VLMs), their applications in personalized recommendation have gained increasing attention. Rec-GPT4V (Liu et al., 2024) proposed the Visual-Summary Thought (VST) mechanism, which integrates image summaries with textual titles, consistently improving the performance of multimodal sequential recommendation (MMSR). Zhou et al. (2025) introduced MSR-Bench to explore how vision-language models can be applied to recommendation using different strategies. I-LLMRec (Kim et al., 2025) directly employed images as item representations to reduce token usage while preserving semantic information, enhancing robustness against textual noise. Towards a foundation model for multi-task learning with vision-language models, VIP5 (Geng et al., 2023) incorporates visual information with multiple instruction templates across different tasks. Similarly, UniMP (Wei et al., 2024a) incorporated images, texts, and cross-modal histories into a unified framework to tackle diverse personalization tasks. These studies highlight the potential of VLMs to enhance multimodal personalized recommendation, However, existing VLM-based recommender systems struggle to capture the personalized visual features since they simply rely on the general-purpose vision-language models that does not have user preference, making them ineffective at modeling users' visual preferences.

**Contrastive Learning for Recommendation.** Contrastive learning (Chen et al., 2020; Xie et al., 2022) has emerged as a powerful tool for learning representation and could also enhance generalization in recommendation systems and address data sparsity. Traditional approaches focus on user–item interactions, whereas recent methods leverage auxiliary information, such as reviews, to enrich latent representations. For example, ReHCL (Wang et al., 2024) constructs topic-based and semantic graphs from user and item reviews, strengthens node representations through cross-view contrastive learning by incorporating rating–review correlations. Similarly, the ReCAFR (Dong et al., 2025) aligns user–item–review representations in a shared embedding space, enabling robust learning even when review data is limited. These studies demonstrate that contrastive learning effectively utilizes multi-layer information (review, user, item), ensures alignment between users and items. However, existing VLM-based recommenders struggle to model personalized visual features, as they rely on general-purpose models without user preferences. To address this, we propose a dual contrastive learning that learns personalized visual representations and aligns them with textual information.

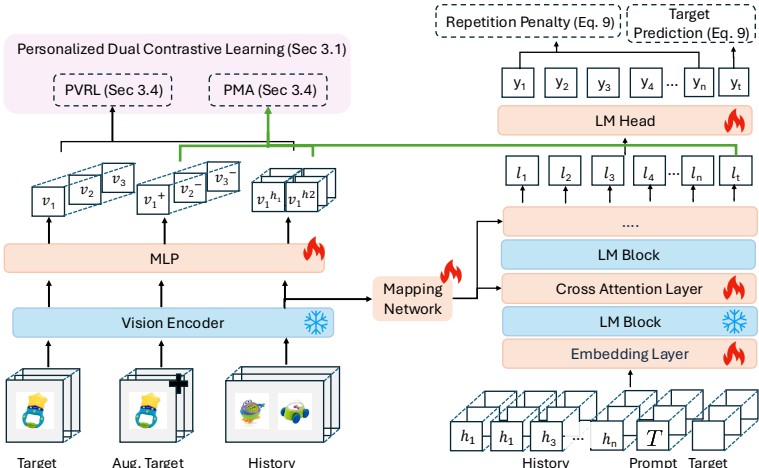

Figure 2: Overview of PerVRA, which enhances personalized multimodal recommendation through two key components: Personalized Visual Representation Learning (PVRL), which refines visual features for personalization, and Personalized Multimodal Alignment (PMA), which guides textual representations in the LLM using personalized visual signals. A dual contrastive learning strategy jointly captures user-level coherence and recommendation-level discriminability by treating historical items in a dual role. The final objective combines these contrastive losses with task-specific recommendation and repetition penalty loss.

# 3 METHOD

We propose a Personalized Visual Representation Alignment (PerVRA) framework that leverages users' historical interactions to learn personalized multimodal representations for recommendation illustrated in Fig. 2. Sec. 3.1 provides preliminaries on applying VLMs to personalized multimodal recommendation. PerVRA employs dual contrastive learning objectives, which jointly capture user-level coherence and recommendation-level discriminability in Sec. 3.2. These objectives are applied within the representation spaces of VLMs through two components: personalized visual representation learning in Sec. 3.3 and personalized multimodal alignment in Sec. 3.4. By integrating these components, PerVRA reshapes the multimodal embedding space to more effectively capture individual user preferences.

## 3.1 PRELIMINARY: RECOMMENDATION WITH HETEROGENEOUS USER HISTORY INFORMATION

PerVRA is built on top of the state-of-the-art multimodal recommendation model UniMP (Wei et al., 2024a), which we select for its strong baseline performance and effectiveness in diverse personalized multimodal recommendation tasks using heterogeneous user history information.

User interaction histories in personalized recommendation extend beyond simple item IDs and often include multimodal attributes such as product images, textual descriptions, categories, and prices. To capture leverage multimodal information, we represent each item with an attribute dictionary that encodes heterogeneous key–value pairs together with its visual feature. These attributes are expressed in natural language, making them directly compatible with language models. We then linearize the dictionary into a sequence, inserting special tokens like [IMG] to indicate visual inputs and [EOC] to signal the end of each item's representation as follows:

$$h_i = \{[\text{IMG}], k_1, v_1, k_2, v_2, \ldots, k_m, v_m, [\text{EOC}]\}, \tag{1}$$

Here, $k$ refers to the attribute type (*e.g.*, category), while $v$ specifies its associated value (*e.g.*, Laptop) from an item $i$. To represent a user's ($u_j$) interaction history, we concatenate all item sequences into a single unified sequence, referred to as the user sentence:

$$u_n = \{[\text{CLS}], h_1, h_2, \ldots, h_n\}, \tag{2}$$

By concatenating these item sequences into a user sentence prefixed with [CLS], the model can process full multimodal histories instead of relying only on textual or ID-based inputs.

With $u_n$, we make an appropriate prompt with an instruction tuning for each recommendation task. For example, in sequential recommendation, we use the task instruction $T$, "Given $u_n$, what is the next item recommended to the user? " (see details in Appendix Sec. C).

## 3.2 DUAL CONTRASTIVE LEARNING FOR PERSONALIZED REPRESENTATION

To optimize personalized representations in VLMs, we introduce a dual personalized contrastive learning tailored for recommendation. The goal is to simultaneously capture user-level preferences (*i.e.*, user-oriented contrastive learning) and recommendation-level discriminability (*i.e.*, recommendation-oriented contrastive learning) within the multimodal embedding space. Specifically, our approach consists of two complementary objectives:

**User-oriented Contrastive Learning (UCL).** This objective treats all items from a user's historical interactions as positives, encouraging them to be embedded close to each other in the latent space. In doing so, the model learns consistent visual representations that reflect user-specific preferences such as styles, colors, categories, or shapes.

Formally, let $z$ denote the feature representation of an image or text from each model, $z^+$ an augmented version of $z$ (via dropout), $z^-$ the representation of a negative item sampled from in-batch examples, and $z^h$ the set of features corresponding to a user's historical items. We compute cosine similarities as:

$$s_{pos} = \cos(z, z^+), \quad s_{neg} = \cos(z, z^-), \quad s_{hist} = \cos(z, z^h) \tag{3}$$

To incorporate user history into the representation space, we treat $z^h$ as additional positives alongside $z^+$. This enforces the target item to be closer to both its augmentations and the $N$ historical items:

$$\mathcal{L}_{\text{UCL}}(z, z^+, z^h, z^-) = -\log \frac{\exp\left(s_{pos}/\tau\right) + \sum_{n=1}^{N} \exp\left(s_{hist,n}/\tau\right)}{\exp\left(s_{pos}/\tau\right) + \sum_{n=1}^{N} \exp\left(s_{hist,n}/\tau\right) + \sum_{b=1}^{B} \exp\left(s_{neg,b}/\tau\right)} \tag{4}$$

**Recommendation-oriented Contrastive Learning (RCL).** Complementary to UCL, this objective emphasizes the discriminability of target items by contrasting them against the user's historical items. Unlike UCL, here history items are treated as challenging negatives, forcing the model to distinguish the target recommendation from items the user has already interacted with, while still aligning with the ground-truth positive:

$$\mathcal{L}_{\text{RCL}}(z, z^+, z^h) = -\log \frac{\exp\left(s_{pos}/\tau\right)}{\exp\left(s_{pos}/\tau\right) + \sum_{n=1}^{N} \exp\left(s_{hist}/\tau\right)} \tag{5}$$

The core idea of dual contrastive learning is to leverage a user's historical items in a **dual role**. On one hand, they are treated as **positives**, enabling the model to capture user-specific preferences such as colors, categories, and shapes. On the other hand, they are treated as **negatives**, preventing the model from simply reproducing past choices and encouraging it to focus on recommendation-oriented distinctions between new candidate items and historical interactions.

Together, these dual contrastive learning objectives enable the model to learn personalized visual features (Sec. 3.3) that capture both consistency across a user's history and discriminability for new recommendations. The enhanced visual representations are then used to guide textual features in the LLM, achieving effective personalized multimodal alignment (Sec. 3.4).

## 3.3 PERSONALIZED VISUAL REPRESENTATION LEARNING

Previous VLMs for multimodal recommendation depend on frozen visual features trained for broad visual perception, which limits their ability to capture user-specific preferences in recommendation tasks. To overcome this limitation, we propose **P**ersonalized **V**isual **R**epresentation **L**earning (PVRL), a module that refines visual features based on each user's historical interactions.

Let $v$ denote the image feature of the current target item, obtained from the image encoder and further projected through an MLP layer. Let $v^+$ be its augmented version generated via dropout, $v^-$ the image feature of a negative item sampled from in-batch candidates, and $v^h$ the image features

of the user's historical items. With these visual features, we perform dual contrastive learning as follows:

$$\mathcal{L}_{\text{PVRL}} = \lambda_1 \mathcal{L}_{\text{UCL}}^v(v, v^+, v^h, v^-) + \lambda_2 \mathcal{L}_{\text{RCL}}^v(v, v^+, v^h) \tag{6}$$

This dual formulation reflects the intuition that history items are not only supportive signals of user preference (positives) but also potential distractors that require fine-grained separation (negatives).

### 3.4 Personalized Multimodal Alignment

While the previous module enhances visual representations, this stage focuses on refining LLM representations for recommendation by leveraging dual contrastive learning. Furthermore, VLMs still tend to underutilize visual information in the later LLM layers Yoon et al. (2025), often defaulting to textual dominance in recommendation. To address this limitation, we propose **P**ersonalized **M**ultimodal **A**lignment (PMA), which leverages personalized visual representations to guide the alignment between image and text features, ensuring that visual signals are more effectively preserved and utilized in the final recommendation.

Let $l_t$ represent the latent representation of the target item from the final LLM layers. The contrastive loss is formulated similarly to the PVRL case, but with a different anchor, $l_t$:

$$\mathcal{L}_{\text{PMA}} = \lambda_1 \mathcal{L}_{\text{UCL}}^t(l_t, v^+, v^h, v^-) + \lambda_2 \mathcal{L}_{\text{RCL}}^t(l_t, v^+, v^h) \tag{7}$$

where $\mathcal{L}_{\text{UCL}}^t$ treats visual historical items as positives alongside the target text, and $\mathcal{L}_{\text{RCL}}^t$ treats history as negatives against the target text. Importantly, the history features used here are personalized visual representations, thereby enabling vision to explicitly guide the alignment process in LLM representations. Finally, the objective of PerVRA is defined as follows:

$$\mathcal{L}_{\text{PerVRA}} = \mathcal{L}_{\text{PVRL}} + \mathcal{L}_{\text{PMA}} \tag{8}$$

**Final Objectives.** $\mathcal{L}_{\text{PerVRA}}$ is combined with the following recommendation task losses.

$$\mathcal{L}_{\text{rec}} = \underbrace{-\log p\left(y_{>n} \mid u_{\leq n} = \{u_n, T\}\right)}_{\text{target item prediction}} \underbrace{-\log p\left(u_n = \{h_1, \dots\}\right)}_{\text{history reconstruction}} \underbrace{-\log(1 - p\left(h \mid u_{\leq n} = \{u_n, T\}\right))}_{\text{repetition penalty}} \tag{9}$$

The loss $\mathcal{L}_{rec}$ consists of three components. Wei et al. (2024a) propose the target item prediction term encourages the model to accurately predict the next item given the user's interaction history. The history reconstruction term ensures that the model preserves and reconstructs the user's historical preferences, reinforcing personalized representation learning.

However, we observed that the history reconstruction loss tends to make VLMs predict historical items as target recommendations. To address this, we introduce a **repetition penalty** term that discourages the model from recommending the same items repeatedly, thereby enhancing diversity. Together, these components ensure a balance between accuracy, personalization, and diversity in recommendations. The final training objective for VLMs for recommendation is defined as follows:

$$\mathcal{L} = \mathcal{L}_{\text{rec}} + \mathcal{L}_{\text{PerVRA}} \tag{10}$$

## 4 Experiments

We next present an extensive experimental result and analysis of PerVRA, focusing on the following key research questions:

- (RQ1) Does PerVRA enhance personalized multimodal sequential recommendation, and how well does it generalize across diverse benchmarks?
- (RQ2) Does PerVRA improve the performance on diverse personalized multimodal recommendation tasks?
- (RQ3) Beyond accuracy, does PerVRA provide additional benefits such as robustness to missing modalities?
- (RQ4) How does model performance vary when combining personalized visual representation learning with personalized multimodal alignment?
- (RQ5) How does the embedding space evolve after applying personalized visual representation learning?

4.1 SETUP

**Dataset.** We utilized multiple publicly available datasets to evaluate the personalized recommendation performance of our model. The primary datasets include the Amazon review dataset (Ni et al., 2019), the H&M fashion dataset (García Ling et al., 2022), which emphasizes item images, and, for broader experimentation, the Netflix (Wei et al., 2024b) and Book-Crossing (Möbius, 2023) datasets. For the Amazon dataset, we combined the Baby, Beauty, Clothing, Grocery, Sports, and Toys domains, while single-domain experiments were conducted using only the Beauty domain. The H&M fashion dataset is particularly suitable for experiments leveraging image information due to the importance of item images. Across all datasets, training items were limited to those with at least five interaction records. We provide additional details in Appendix Sec. A.

**Implementation Details.** We implement our framework using UniMP (Wei et al., 2024a) as the backbone. The visual encoder is CLIP's ViT-L (Radford et al., 2021) followed by an MLP, while the LLM is a 3B instruction-tuned large language model (Together.ai, 2023). See Appendix Sec. B for more details.

**Personalized Multimodal Recommendation Tasks.** To comprehensively evaluate the capabilities of our model, we follow UniMP Wei et al. (2024a) and conduct experiments across diverse recommendation tasks: personalized sequential recommendation, multimodal search, multimodal selection, explanation generation, and preference prediction. These tasks collectively assess the model's ability to capture user preferences, model temporal dynamics, generate explanations, align recommendations with user choices, and return relevant results to queries. Detailed task descriptions are provided in the Appendix Sec. C.

Table 1: Performance Comparison on Sequential Recommendation on Amazon Reviews.

| | HR@3 | NDCG@3 | MRR@3 | HR@5 | NDCG@5 | MRR@5 |
|---|---|---|---|---|---|---|
| MF | 0.010 | 0.008 | 0.007 | 0.016 | 0.009 | 0.008 |
| MACR | 0.011 | 0.008 | 0.007 | 0.017 | 0.010 | 0.009 |
| LightGCN | 0.014 | 0.010 | 0.008 | 0.020 | 0.013 | 0.010 |
| UltraGCN | 0.015 | 0.011 | 0.009 | 0.021 | 0.013 | 0.010 |
| HGN | 0.016 | 0.011 | 0.011 | 0.023 | 0.015 | 0.011 |
| GRU4Rec | 0.013 | 0.010 | 0.008 | 0.020 | 0.012 | 0.009 |
| SASRec | 0.018 | 0.012 | 0.010 | 0.027 | 0.017 | 0.012 |
| $S^3$-Rec | 0.020 | 0.014 | 0.013 | 0.031 | 0.019 | 0.015 |
| BERT4Rec | 0.012 | 0.009 | 0.007 | 0.019 | 0.012 | 0.010 |
| UniSRec | 0.020 | 0.014 | 0.013 | 0.028 | 0.019 | 0.015 |
| VBPR | 0.011 | 0.008 | 0.007 | 0.018 | 0.010 | 0.008 |
| CausalRec | 0.014 | 0.010 | 0.008 | 0.022 | 0.014 | 0.012 |
| MMGCL | 0.015 | 0.011 | 0.009 | 0.024 | 0.015 | 0.013 |
| MMSSL | 0.018 | 0.013 | 0.011 | 0.028 | 0.019 | 0.016 |
| P5 | 0.008 | 0.005 | 0.004 | 0.012 | 0.007 | 0.006 |
| VIP5$^+$ | 0.017 | 0.012 | 0.010 | 0.026 | 0.016 | 0.012 |
| UniMP | 0.032 | 0.020 | 0.016 | 0.057 | 0.030 | 0.021 |
| Ours | **0.058** | **0.042** | **0.037** | **0.076** | **0.049** | **0.041** |

Table 2: Performance Comparison on Sequential Recommendation across Benchmarks.

| | Netflix | | H&M | | Book | | Amazon Beauty | |
|---|---|---|---|---|---|---|---|---|
| | HR@5 | NDCG@5 | HR@5 | NDCG@5 | HR@5 | NDCG@5 | HR@5 | NDCG@5 |
| P5 | 0.074 | 0.043 | 0.010 | 0.006 | - | - | - | - |
| VIP5$^+$ | 0.093 | 0.058 | 0.012 | 0.011 | - | - | - | - |
| UniMP | 0.163 | 0.115 | 0.077 | 0.055 | 0.119 | 0.072 | 0.062 | 0.037 |
| Ours | **0.173** | **0.121** | **0.087** | **0.060** | **0.130** | **0.084** | **0.130** | **0.115** |

4.2 PERFORMANCE COMPARISON

**Results on sequential recommendation across datasets (RQ1).** Tables 1 and 2 summarize performance comparisons on sequential recommendation across benchmarks. Notably, on Amazon

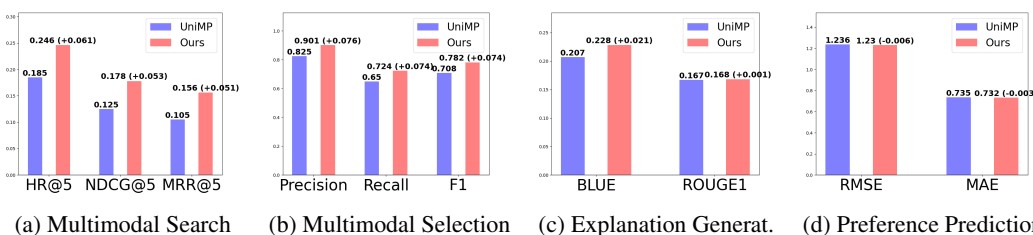

(a) Multimodal Search    (b) Multimodal Selection    (c) Explanation Generat.    (d) Preference Prediction

Figure 3: Performance Comparison for Personalized Multimodal Tasks on Amazon Reviews.

Reviews (Table 1), PerVRA achieves an NDCG@5 of 0.049, yielding more than 30% relative improvement over the strongest baseline UniMP. Similarly, on diverse benchmarks including Netflix, H&M, Book, and Amazon Beauty (Table 2), PerVRA consistently surpasses the baselines, achieving the highest performance across all datasets. These results confirm the effectiveness and generalizability of our personalized visual representation alignment in sequential recommendation tasks.

**Results on diverse recommendation tasks (RQ2).** Fig. 3 shows performance comparisons on multi-task recommendation with UniMP, covering multimodal search, selection, explanation generation, and preference prediction. Our method achieves substantial gains in multimodal search and selection—tasks that rely heavily on multimodal signals—while maintaining competitive performance on explanation generation and preference prediction, which primarily depend on textual reviews. We provide full results in the Appendix.

In conclusion, PerVRA enables the model to fully utilize visual cues, yielding notable performance gains over the baseline across multimodal recommendation tasks without additional inference costs.

## 4.3 Additional Benefit of Multimodal Alignment: Robustness to Missing Modalities (RQ3)

Table 3: Robustness Evaluation under Missing Modalities

(a) Performance on H&M

| Evaluation | UniMP | | Ours | |
|---|---|---|---|---|
| | HR@5 | NDCG@5 | HR@5 | NDCG@5 |
| multimodal | 0.064 | 0.043 | 0.078 | 0.054 |
| text-only | 0.054 | 0.033 | 0.079 | 0.053 |

(b) Performance on Amazon Reviews

| Evaluation | UniMP | | Ours | |
|---|---|---|---|---|
| | HR@5 | NDCG@5 | HR@5 | NDCG@5 |
| multimodal | 0.058 | 0.028 | 0.073 | 0.042 |
| text-only | 0.047 | 0.026 | 0.074 | 0.045 |

Up to this point, models have been trained under multimodal settings where paired images and text are always available. However, in real-world scenarios, images may be missing or unavailable. It is therefore crucial to ensure the model can effectively handle such cases. In this section, we investigate the robustness of both baselines and our proposed method under this setting.

To further investigate the robustness of our model, we conducted experiments under a setting where only half of the image information was provided during training. For evaluation, we considered two conditions: (1) Multimodal evaluation, where full image information is available, and (2) text-only evaluation, where images are excluded. In missing-modality scenarios, it is desirable for models to maintain performance comparable to the full-modality setting.

In Table 3, results on the H&M dataset—where visual information is critical for recommendation—show that baseline models suffer a substantial performance drop when evaluated with text-only input compared to multimodal input. In contrast, our model exhibits a much smaller drop, with only a marginal gap between text-only and multimodal evaluations, demonstrating stable performance across modalities. Similar trends are observed on the Amazon Reviews dataset, confirming the generalizability of our findings.

Overall, we attribute this robustness to our model's ability to leverage vision-language alignment via PMA, which enables textual data to encode visual cues and sustain strong performance even under incomplete visual information.

## 4.4 ABLATION STUDIES (RQ4)

Table 4 presents the ablation study of each component in PerVRA. We first observe that replacing our dual personalized contrastive learning with a standard contrastive objective (w/ InfoNCE Chen et al. (2020)) leads to a sharp drop in performance. Removing PVRL also results in a clear performance degradation, showing the importance of refining personalized vision features. Eliminating either RCL or UCL causes notable decreases, especially in NDCG@5 and MRR@5, indicating that

Table 4: Ablation Studies.

|  | HR@5 | NDCG@5 | MRR@5 |
|---|---|---|---|
| w/ InfoNCE | 0.066 | 0.038 | 0.029 |
| w/o PVRL | 0.065 | 0.039 | 0.031 |
| w/o RCL | 0.066 | 0.045 | 0.039 |
| w/o UCL | 0.073 | 0.046 | 0.037 |
| w/o Repetition Penalty | 0.068 | 0.042 | 0.035 |
| Ours | **0.076** | **0.049** | **0.041** |

both recommendation-oriented and user-oriented contrastive learning contribute complementary benefits. Finally, removing the repetition penalty consistently lowers performance, highlighting its role in discouraging redundant recommendations. These results demonstrate that each component of PerVRA plays an essential role, and their integration leads to the overall superior performance reported.

## 4.5 REPRESENTATION SPACE ANALYSIS WITH DUAL PERSONALIZED CONTRASTIVE LEARNING (RQ5)

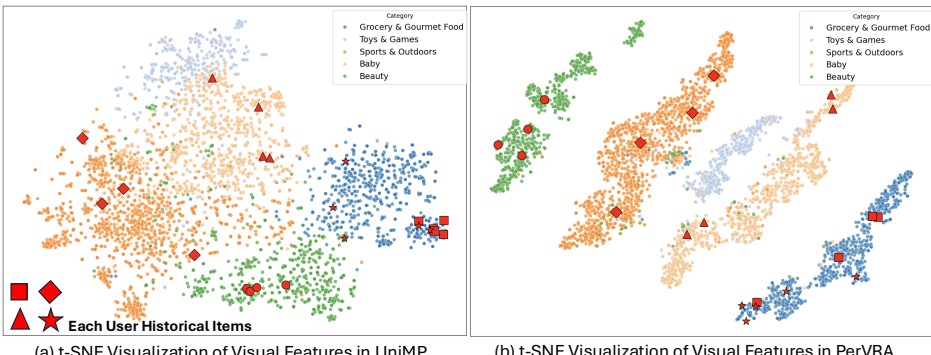

(a) t-SNE Visualization of Visual Features in UniMP     (b) t-SNE Visualization of Visual Features in PerVRA

Figure 4: Visual Feature Visualization of UniMP and PerVRA.

In Figure 1, we analyzed the hidden space representations $(l_1, l_2, \ldots)$ in VLM recommenders, showing that PerVRA yields more structured latent clusters compared to UniMP. To further examine how visual signals contribute to this alignment, we now present t-SNE visualizations of the raw vision features in Figure 4.

In UniMP (Figure 4-(a)), visual features remain highly entangled across categories. In contrast, PerVRA (Figure 4-(b)) produces well-separated clusters aligned with item categories. This shows that the enhanced visual features in PerVRA further improve the quality of the hidden latent space through PMA, ultimately enabling more effective personalized multimodal recommendation.

## 4.6 EFFECT OF VISION ENCODER TRAINING IN PERSONALIZED MULTIMODAL RECOMMENDATION

To investigate whether fine-tuning the vision encoder can enhance personalized recommendation performance, we conduct an ablation study comparing different training strategies for the visual encoder. Table 5 summarizes the results across benchmarks.

We compare three variants: (1) the baseline UniMP with a frozen vision encoder, (2) UniMP with a fully trainable vision encoder, and (3) UniMP with LoRA-based vision encoder adaptation to preserve pre-trained representations while enabling limited tuning. As shown in Table 2, directly fine-tuning the vision encoder leads to degraded performance, likely because the original discriminative visual representation is distorted. The LoRA-based approach partially alleviates this issue but still falls short of our proposed method, which achieves the best overall results.

Our method outperforms all baselines without fine-tuning the vision encoder by introducing a Per-VRA mechanism. This design allows the model to adapt visual embeddings to user preferences while maintaining the integrity of pre-trained visual knowledge.

Table 5: Performance Comparison on Sequential Recommendation across Benchmarks.

|  | HR@5 | NDCG@5 | MRR@5 |
|---|---|---|---|
| UniMP | 0.057 | 0.030 | 0.021 |
| + Vision Training | 0.046 | 0.029 | 0.018 |
| + LoRA Vision Training | 0.052 | 0.029 | 0.019 |
| Ours | **0.076** | **0.049** | **0.041** |

### 4.7 GENERALIZABILITY ANALYSIS: APPLICABILITY OF PERVRA TO DIVERSE VLM BACKBONES

While the primary implementation of the PerVRA framework utilizes UniMP—which adopts a Flamingo-based architecture-we explore other VLM architectures with Qwen2.5VL, a VLM characterized by a fundamentally different structural design and pre-training alignment properties compared to the Flamingo-based UniMP.

As observed in Table 6, applying PerVRA to Qwen2.5VL results in consistent performance improvements across all metrics.

Table 6: Performance Comparison with Qwen2.5VL.

| Method | HR@5 | NDCG@5 | MRR@5 |
|---|---|---|---|
| Qwen2.5VL + UniMP | 0.037 | 0.019 | 0.015 |
| Qwen2.5VL + PerVRA | **0.043** | **0.028** | **0.017** |

### 4.8 ROBUSTNESS ACROSS MODEL SIZES

We conduct an additional experiment applying PerVRA to a 1B parameter LLM (mosaaicml, 2023), which is significantly smaller than the 3B model used in the main paper. As shown in the additional results (Table 7), PerVRA achieves remarkable performance gains over the UniMP baseline even on the 1B model:

These relative improvements are consistent with the gains observed in the 3B model experiments. This empirically proves that PerVRA's effectiveness is driven by its personalized alignment mechanism rather than model scale.

Table 7: Performance Comparison with 1B LLM.

| Model | HR@5 | NDCG@5 | MRR@5 |
|---|---|---|---|
| UniMP-1B | 0.040 | 0.023 | 0.018 |
| PerVRA-1B | **0.069** | **0.042** | **0.034** |

## 5 CONCLUSION

In summary, although existing VLM-based recommenders benefit from powerful vision–language pretraining, they fall short in modeling fine-grained user preferences and fully utilizing visual signals from user histories, largely due to frozen vision encoders trained primarily for perception. To overcome these limitations, we propose PerVRA, which refines personalized visual features and aligns them with textual embeddings via dual personalized contrastive learning, ensuring their effective use in multimodal recommendation. By reshaping the multimodal space toward recommendation objectives, PerVRA provides more accurate, user-centered recommendations without incurring additional inference overhead.

ETHICS STATEMENT

During the preparation of this manuscript, the author utilized LLMs to assist with improving readability. The author carefully reviewed and revised all content generated with this tool and assumes full responsibility for the accuracy and integrity of the final version.

REPRODUCIBILITY STATEMENT

Our model and data are based on the publicly available model Wei et al. (2024a). Additional statistics and details of the dataset are presented in the Appendix. We will release our code upon acceptance.

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

## A  DATA STATISTICS

Table 8: Dataset Statistics.

| Domain | SubDomain | Users | Items | Interactions |
|---|---|---|---|---|
| Amazon | Baby | 19,822 | 7,776 | 163,856 |
| | Beauty | 25,837 | 16,893 | 227,920 |
| | Clothing | 58,197 | 44,310 | 422,474 |
| | Grocery | 16,318 | 11,581 | 165,893 |
| | Sports | 40,358 | 24,766 | 334,238 |
| | Toys | 24,314 | 18,906 | 209,281 |
| Book | | 278,858 | 271,379 | 1,149,780 |
| H&M | | 1,362,281 | 45,875 | 31,788,324 |
| Netflix | | 13,187 | 17,366 | 68,933 |

We utilize multiple public datasets to comprehensively evaluate recommendation performance across different domains. The Amazon dataset consists of six subdomains (Baby, Beauty, Clothing, Grocery, Sports, and Toys), each exhibiting diverse distributions of users and items.

The Book dataset contains a large number of items (over 270K). The H&M dataset provides large-scale interaction data collected from a real-world fashion retail environment, enabling evaluation under an industrial-scale setting. Lastly, the Netflix dataset serves as a representative benchmark for video content recommendation, with a relatively balanced distribution of user–item interactions.

Table 8 summarizes the number of users, items, and interactions for each dataset, illustrating that our experiments cover datasets with diverse scales and characteristics to ensure generalizability.

## B  IMPLEMENTATION DETAILS

In this study, the model was trained with a learning rate of 2e-4 and a batch size of 8. Gradient accumulation steps were set to 2 to facilitate effective large-batch training. The model was trained for 10 epochs, and a cosine decay schedule was applied to the learning rate. A warmup ratio of 0.01 was used at the beginning of training

Training was conducted on four A6000 GPUs using DeepSpeed in combination with Hugging Face Accelerate, with Zero Stage 2 optimization applied. Mixed precision (bf16) was employed to improve memory efficiency and accelerate training.

For the loss functions, the weighting parameters $\lambda_1$ and $\lambda_2$ in Equations (6) and (7) were set to 1.5 and 0.5, respectively. The temperature parameter $\tau$ used in Equations (4) and (5) was fixed at 0.07. The MLP shown in Figure 2 was implemented as a 2-layer MLP with a hidden layer size of 1024. In addition, the weighting coefficients for $\mathcal{L}_{\text{PVRL}}$ and $\mathcal{L}_{\text{PMA}}$ in Equation (8) were both set to 0.3, which was found to yield the best performance in our experiments.

For the visual modality, we also adopt the original product images as inputs. The textual side consists of item metadata such as category, title, price, ID, and brand. Following UniMP(Wei et al., 2024a), each item in the corpus is assigned a unique token in the vocabulary table. For evaluation, we construct the user history using the last five interactions. In both the search and recommendation tasks, beam search with a size of 10 is employed to generate candidate items.

Table 9: Sensitivity Analysis on (a) the Number of MLP Layers, (b) Loss-Weight Parameters, and (c) Temperature on the Amazon Dataset Under the Sequential Recommendation Task.

**(a) Effect of MLP Depth**

| #Layers | HR@5 | NDCG@5 | MRR@5 |
|---|---|---|---|
| 1-Layer | 0.062 | 0.038 | 0.031 |
| 2-Layer (ours) | **0.076** | **0.049** | **0.041** |
| 3-Layer | 0.060 | 0.035 | 0.028 |

**(b) Effect of Loss Weight Parameters**

| $\lambda_1$ | $\lambda_2$ | HR@5 | NDCG@5 | MRR@5 |
|---|---|---|---|---|
| 1.0 | 1.0 | 0.065 | 0.041 | 0.034 |
| 1.5 | 0.5 | **0.076** | **0.049** | **0.041** |
| 0.5 | 1.5 | 0.052 | 0.037 | 0.032 |
| 2.0 | 2.0 | 0.045 | 0.034 | 0.031 |

**(c) Effect of temperature**

| $\tau$ | HR@5 | NDCG@5 | MRR@5 |
|---|---|---|---|
| 0.05 | 0.051 | 0.035 | 0.030 |
| 0.07 | **0.076** | **0.049** | **0.041** |
| 0.1 | 0.073 | 0.047 | 0.038 |

## C  TASK DESCRIPTIONS

To evaluate the diverse capabilities of the model, we employed five tasks defined in UniMP(Wei et al., 2024a). Each task reflects a core function of recommendation systems and is designed to assess how effectively the model leverages personalized information during training and evaluation.

- Sequential Recommendation task aims to predict the next item based on a user's historical interaction sequence. This task allows the model to learn users' evolving preferences and behavioral patterns, highlighting the importance of temporal interaction information.

  *Prompt:* "recommendation: What is the next item recommended to the user?"

- Preference Prediction: The preference prediction task involves estimating a user's preference for a specific item, using either explicit ratings or implicit feedback. This task evaluates how accurately the model captures individual user tastes.

  *Prompt:* "What is the rating and explanation for the item?"

- Explanation Generation task requires the model to produce natural language explanations for recommended items. This task assesses the model's ability to generate understandable explanations and provide personalized reasoning behind recommendations.

  *Prompt:* "What is the rating and explanation for the item?"

- Selection task predicts which item a user is most likely to select from a given set of candidates. It focuses on evaluating the alignment between the model's recommendations and actual user choice behavior.

  *Prompt:* "selection: Can you select the suitable item from above for the user?"

- Search task evaluates the model's ability to return relevant items in response to user queries. It measures how well the model understands search intent and provides personalized results.

  *Prompt:* "search: Query: {query} What is the related item ID to the query based on the history?"

## D  SENSITIVITY ANALYSIS

We conduct a sensitivity analysis to examine how the architectural and loss design choices affect the performance of our PerVRA framework. Specifically, we vary (a) the number of MLP layers used

for projecting image features and (b) the weighting coefficients $\lambda_1$ and $\lambda_2$ that control the contributions of the user-oriented contrastive loss ($\mathcal{L}_{\text{UCL}}$) and the recommendation-oriented contrastive loss ($\mathcal{L}_{\text{RCL}}$), and (c) the temperature parameter $\tau$ used in Equation (4) and Equation 5.

**Effect of MLP Depth.** Table 9(a) shows the effect of varying the depth of the MLP projection used for refining visual representations. The 2-layer configuration achieves the best results across all metrics, surpassing both the 1-layer and 3-layer variants. A moderate projection depth provides sufficient capacity to encode user-specific visual semantics—such as colors, categories, and styles—while preserving stable multimodal alignment. In contrast, a 1-layer MLP underfits the complex relationships between user history and target items, whereas a deeper 3-layer model introduces excessive nonlinearity, which can distort the alignment between modalities and hinder generalization.

**Effect of Loss Weight Parameters.** Table 9(b) reports the performance when adjusting $\lambda_1$ and $\lambda_2$, which balance $\mathcal{L}_{\text{UCL}}$ and $\mathcal{L}_{\text{RCL}}$. We observe that setting $\lambda_1 = 1.5$ and $\lambda_2 = 0.5$ yields the best performance. This indicates that emphasizing $\mathcal{L}_{\text{UCL}}$—which aligns the target and historical items within a shared user preference space—helps the model build a coherent user-level representation. At the same time, a smaller weight on $\mathcal{L}_{\text{RCL}}$ ensures sufficient fine-grained discrimination between the target and the user's past interactions without breaking the overall preference consistency. When $\lambda_2$ becomes dominant, the model focuses excessively on separating history items, leading to over-differentiation within the user preference manifold and, consequently, degraded recommendation performance.

Overall, these results highlight the importance of maintaining a balanced interplay between user preference alignment (UCL) and recommendation-level discrimination (RCL) in achieving effective personalized multimodal representations.

**Effect of Temperature Parameter.** Table 9(c) presents the performance variations with respect to the temperature parameter $\tau$ used in our dual contrastive learning objectives (Eq. (4) and Eq. (5)). The results indicate that setting $\tau = 0.07$ achieves superior performance across all metrics, yielding an HR@5 of 0.076 and an NDCG@5 of 0.049. We observe a significant performance degradation when $\tau$ is set too low (i.e., 0.05). This is likely because an excessively low temperature sharpens the distribution, causing the model to over-focus on hard negatives or leading to optimization instability within the embedding space. Conversely, increasing $\tau$ to 0.1 results in better performance than 0.05 but falls short of the peak performance observed at 0.07. This suggests that higher temperatures overly smooth the distribution, thereby weakening the model's discriminability in distinguishing fine-grained differences between positive pairs (user history and target items) and negatives. Consequently, $\tau = 0.07$ serves as the optimal balance point, effectively maintaining user preference consistency while maximizing discriminability for recommendation.

## E    FULL RESULTS ON DIVERSE PERSONALIZED MULTIMODAL RECOMMENDATION TASKS

Table 10: Performance Comparison for Personalized Multimodal Tasks on Amazon Reviews.

|  | HR@3 | NDCG@3 | MRR@3 | HR@5 | NDCG@5 | MRR@5 |
|---|---|---|---|---|---|---|
| UniMP | 0.135 | 0.105 | 0.094 | 0.185 | 0.125 | 0.105 |
| Ours | **0.196** | **0.158** | **0.145** | **0.246** | **0.178** | **0.156** |

(a) Personalized Multimodal Search.

|  | recall | precision | f1 |
|---|---|---|---|
| UniMP | 0.65 | 0.825 | 0.708 |
| Ours | **0.724** | **0.901** | **0.782** |

|  | MAE | RMSE |
|---|---|---|
| UniMP | 0.735 | 1.236 |
| Ours | **0.732** | **1.23** |

(b) Personalized Multimodal Selection.    (c) Personalized Preference Prediction.

|  | BLEU | Rouge1 | Rouge2 | RougeL | METEOR | BERTSCORE |
|---|---|---|---|---|---|---|
| UniMP$^+$ | 0.207 | 0.167 | 0.025 | 0.140 | 0.122 | 0.862 |
| Ours | **0.228** | **0.168** | **0.027** | **0.141** | 0.122 | **0.863** |

(d) Personalized Explanation Generation.

To complement the results presented in Figure 3, this appendix provides the complete experimental results for all personalized multimodal recommendation tasks evaluated on the Amazon Reviews dataset in Table 10. Our evaluation covers four key subtasks—**multimodal search**, **multimodal selection**, **preference prediction**, and **explanation generation**—each highlighting different aspects of multimodal reasoning and personalization.

Our method consistently outperforms UniMP across a variety of personalized multimodal tasks. In search tasks (Table 10a), it achieves higher ranking-based metrics (HR@3/5, NDCG@3/5, and MRR@3/5), demonstrating strong ability to retrieve user-preferred items using both textual and visual signals. For multimodal selection (Table 10b), our model surpasses UniMP in recall, precision, and F1-score, indicating improved discriminative understanding of user-specific multimodal cues. In preference prediction (Table 10c), it attains slightly lower MAE and RMSE, reflecting more accurate modeling of user preferences from multimodal reviews. Finally, in explanation generation (Table 10d), our approach outperforms UniMP across most text generation metrics (BLEU, Rouge, BERTSCORE) and achieves comparable performance in METEOR, suggesting better alignment between generated explanations and user-specific multimodal contexts.

Overall, these detailed results confirm that our proposed method maintains balanced performance across diverse multimodal recommendation tasks, achieving notable gains particularly in search and selection tasks that rely heavily on multimodal representations.

## F ROBUSTNESS ANALYSIS: CONSISTENT PERFORMANCE ACROSS MULTIPLE SEEDS

We conduct additional experiments using five random seeds. Table 11 demonstrates that our proposed PerVRA framework achieves statistically significant and stable improvements over the UniMP baseline. The analysis clearly confirms that the reported performance is not dependent on any particular seed.

Table 11: Comparison of Mean $\pm$ Std Across 5 Random Seeds.

|  | UniMP (Baseline) Mean $\pm$ Std | PerVRA (Ours) Mean $\pm$ Std |
|---|---|---|
| HR@5 | $0.0452 \pm 0.0167$ | $0.0673 \pm 0.0068$ |
| NDCG@5 | $0.0249 \pm 0.0092$ | $0.0433 \pm 0.0048$ |
| MRR@5 | $0.0184 \pm 0.0070$ | $0.0356 \pm 0.0045$ |

## G COMPUTATION EFFICIENCY ANALYSIS

Although PerVRA introduces four additional contrastive loss terms, the computational cost does not increase fourfold because the User-oriented (UCL) and Recommendation-oriented (RCL) objectives share the same similarity matrices within each modality. Consequently, the additional complexity is limited to approximately $\mathcal{O}((BH + B^2)D)$, where $B$ denotes the number of negative items (batch size), $H$ represents the length of user history, and $D$ is the feature dimension. In our experiments, we adopted specific values of $H = 5$, $B = 8$, and $D = 2560$, keeping this overhead minimal. Furthermore, the added 2-layer MLP for visual projection is lightweight compared to the LLM backbone. Crucially, these auxiliary modules are applied only during training and are discarded at inference, ensuring that PerVRA incurs no additional computational cost during deployment.

Empirically, we validated this efficiency by measuring wall-clock training times on $4 \times$ NVIDIA A6000 GPUs over 10 epochs. PerVRA required 1 hour 25 minutes compared to the baseline UniMP's 1 hour 18 minutes, representing a marginal increase of approximately 9.0%. Given the significant performance gains observed across diverse tasks, we conclude that this slight increase in training cost is highly justifiable relative to the substantial improvements in recommendation accuracy.

# H    QUALITATIVE RESULTS

To further analyze the recommendation performance of our model, we present qualitative results on the H&M and Amazon datasets in Figures 5a and 5b, respectively.

In each figure, the first row shows the user's history items (the first five items) and the target item (the sixth item). Rows 2 and 3 illustrate the top-10 recommendation results from the baseline model, presented as two rows of five items each. Rows 4 and 5 display the top-10 recommendation results from our model, arranged in two rows of five items per row.

From the visualizations, it can be observed that our model consistently recommends items that are more visually and semantically aligned with the user's history and target items compared to the baseline. For instance, in Figure 5a (H&M dataset), the items recommended by our model capture finer details of style and category preferences demonstrated in the history, while the baseline recommendations appear less relevant. A similar pattern is observed in Figure 5b (Amazon dataset), where our model's recommendations closely match the attributes of the target item, demonstrating improved personalization and contextual understanding. We note that Figure 5c and Figure 5d represent miss cases. In Figure 5c, the user's history consists mostly of single-color fashion items, whereas the target item features more vibrant and diverse colors. In this case, the visual information may have been less helpful for making the recommendation.

These qualitative results confirm that our approach can effectively leverage user history to generate more accurate and contextually appropriate recommendations compared to baseline methods.

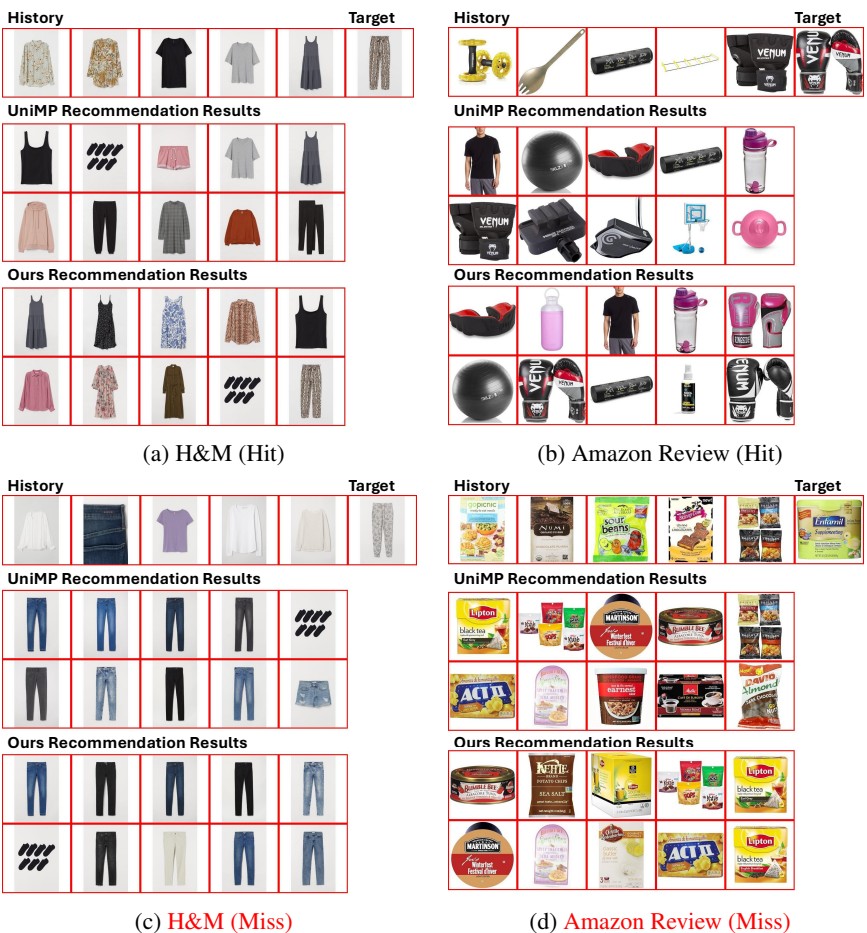

(a) H&M (Hit)       (b) Amazon Review (Hit)

(c) H&M (Miss)       (d) Amazon Review (Miss)

Figure 5: Qualitative Results.

