# OpenReview forum: "Personalized Visual Representation Alignment for Generative Multimodal Recommendation"
_ICLR.cc/2026/Conference — Submitted to ICLR 2026_

### Official Review · Reviewer_zYZi · 2025-10-30

**Soundness:** 3
**Presentation:** 3
**Contribution:** 2
**Rating:** 4
**Confidence:** 4

**Summary:**

This paper proposes the PerVRA framework for VLM-based recommendation systems, employing dual contrastive learning to learn personalized visual representations and align them with textual features. The paper identifies key limitations of existing methods, proposes reasonable solutions, and validates effectiveness across multiple datasets.

**Strengths:**

S1. Figure 1 provides intuitive visual evidence that text-only model outperforms multimodal variant.

S2. The paper clearly identifies two key issues in VLM-based recommendation systems, the motivation is  convincing.

S3. The PVRL and PMA modules have clear division of labor, optimizing visual space and multimodal alignment respectively.

**Weaknesses:**

W1. The contrastive learning in recommender systems is not novel, and the core dual contrastive learning idea lacks originality.

W2.  No analysis of λ1 and λ2 impact, Table 4 only tests complete removal.

W3. Design of Equation 9 lacks theoretical basis, and it may lead to negative loss.

**Questions:**

Q1. Suggest clarifying why this form is used instead of standard InfoNCE in Eq.4.

Q2. Why not contrast lt with lt+ in Eq.7? What are the advantages of this design?

Q3. The  "PerVRA" and "PerVLA" are used mixed in multiple places, e.g., Equation 8.

Q4. On H&M dataset in Table3, PerVRA's text-only evaluation (HR@5=0.079) is even slightly higher than multimodal evaluation (HR@5=0.078). This contradicts core motivation that "visual information is important"

---

> ### Author Response · Authors · 2025-11-22
> **Response to Reviewer zYZi - 1**
>
> > W1: The contrastive learning in recommender systems is not novel, and the core dual contrastive learning idea lacks originality.
>
> We respectfully note that our Dual Contrastive Learning approach is fundamentally different from traditional contrastive learning (CL) methods used in recommendation systems. While we acknowledge that CL is widely used in this domain, our specific formulation and its application within the Vision-Language Model (VLM) framework are fundamentally distinct. Our novelty does not lie in the use of CL itself, but in how we redesign it to address two inherent limitations of VLMs: the lack of personalization in frozen vision encoders and the “textual dominance” bias in LLMs. We summarize our key contributions below.
>
>
> 1. Novelty in Dual Contrastive Learning
> PerVRA introduces a new mechanism in which the user’s interaction history plays dual roles simultaneously. User-oriented Contrastive Learning treats historical items as positive samples together with the target, strengthening preference coherence by clustering items according to the user's preferences. At the same time, Recommendation-oriented Contrastive Learning treats historical items as negative samples relative to the target, preventing the model from merely memorizing past interactions and guiding it to identify the next potentially preferred item. Through Personalized Visual Representation Learning, this dual mechanism directly mitigates the issue of “lack of personalization in frozen vision encoders.”
>
> 2. Resolving VLM Modality Biases
> Personalized Multimodal Alignment applies Dual Contrastive Learning again to align the LLM’s textual representations with the personalized visual features. This explicitly alleviates the modality bias where the LLM tends to underutilize visual signals.
>
> We further demonstrate that our Dual CL substantially differs from standard CL methods. In the ablation study (Table 4), replacing Dual Contrastive Learning with a standard InfoNCE objective leads to a performance drop (HR@5 decreases from 0.076 to 0.066). Moreover, in the t-SNE visualizations (Figures 1 and 4), standard VLM representations (e.g., UniMP) form entangled clusters, whereas our Dual CL produces distinct clusters aligned with user preferences.
>
> > W2: No analysis of λ1 and λ2 impact, Table 4 only tests complete removal.
>
> Effect of Loss Weights: We analyzed the trade-off between the user-oriented contrastive loss ($\mathcal{L}\_{\text{UCL}}$, weighted by $\lambda_1$) and the recommendation-oriented contrastive loss ($\mathcal{L}_{\text{RCL}}$, weighted by $\lambda_2$).  We have added this sensitivity analysis in Table 9 of the revised manuscript. The experimental results indicate that setting $\lambda_1 = 1.5$ and $\lambda_2 = 0.5$ yields the highest accuracy. This suggests that emphasizing user-level preference alignment (UCL) is crucial for building coherent representations, while a moderate weight on RCL ensures fine-grained discrimination without over-differentiation.
>
> > W3: Design of Equation 9 lacks theoretical basis, and it may lead to negative loss.
>
> We apologize for a typographical error in the presentation of Equation 9 in the original manuscript. We have corrected the formula to accurately reflect our implementation and address the reviewer's concern regarding negative loss.
> 1. Correction of Typo
> The equation in the manuscript contained a notation error. The correct formulation, which was used in our actual implementation, is:
> $$\mathcal{L}_\text{repetition penalty} = -\log(1 - p(h|u_{\le n}))$$
> We have revised the manuscript to include the correct sign and structure.
>
> 2. Mathematical Proof of Non-Negative Loss
> With the corrected formula, the concern regarding negative loss is resolved.
> • Since the probability $p(h)$ lies within $[0, 1]$, the term $(1 - p(h))$ is also in $[0, 1]$.
> • The logarithm of a value in this range is non-positive ($\le 0$).
> • Multiplying by $-1$ ensures the final loss term is strictly non-negative ($\ge 0$).
> 3. Theoretical Insights
> This objective is motivated by unlikelihood training [E]. It serves as a necessary structural constraint to counter "History Bias" by explicitly penalizing the repetition of past items during target prediction, as demonstrated by the performance drop when removing it in our ablation study (Table 4).
>
> [E] Welleck, Sean, et al. "Neural text generation with unlikelihood training." arXiv preprint arXiv:1908.04319 (2019).

---

> ### Author Response · Authors · 2025-11-22
> **Response to Reviewer zYZi - 2**
>
> > Q1: Suggest clarifying why this form is used instead of standard InfoNCE in Eq.4.
>
> We sincerely appreciate the reviewer’s insightful comment regarding the design intent of Equation 4 (User-oriented Contrastive Learning, UCL), which is one of the core methodologies in our paper. As the reviewer noted, the standard InfoNCE loss is widely used in contrastive learning. However, in the context of our study—achieving **personalized recommendation**—there was a compelling reason to adopt a modified formulation instead of the standard InfoNCE. We provide the theoretical rationale and empirical evidence as follows:
>
> The standard InfoNCE and our proposed UCL fundamentally differ in their learning goals.
>
> - Standard InfoNCE: Primarily aims at instance discrimination. That is, an image should be close to its own augmented version while being pushed away from all other samples, including the user’s past interaction history. If the standard InfoNCE were applied directly, the model would attempt to push the target item away from the user’s historical preference items, which conflicts with the goal of modeling a user’s consistent tastes in a recommendation system.
> - Proposed Equation 4 (UCL): Our objective is preference coherence. The user’s historical items are **not negatives** to be pushed away but rather **additional positives** that define the user’s preferences. Therefore, we added the historical item term $\sum \exp(s_{hist,n/τ})$ in the numerator of Equation 4. This ensures that the target item is aligned not only with its own augmented version but also with the user’s entire interaction history, forming user-specific cohesive clusters in the latent space.
>
> **Experiments Result:** The validity of this design is demonstrated in Table 4 of our paper (Ablation Studies). The results show that using the standard InfoNCE loss (w/ InfoNCE) instead of our proposed loss causes the HR@5 performance to drop from 0.076 to 0.066, approximately a 13% decrease. This supports our hypothesis that the standard InfoNCE, by treating a user’s historical items as negatives, actually disrupts the personalized visual preference information.
>
> > Q2: Why not contrast $l_t$ with $l_t^+$ in Eq.7? What are the advantages of this design?
>
> We thank the reviewer for this insightful question regarding the design of our Personalized Multimodal Alignment (PMA) objective in Eq. 7.
>
> The decision to contrast the textual representation $l_t$ (from the LLM) with the visual representations ($v^+$ and $v^h$), rather than with an augmented textual representation $l_t^+$, is a deliberate design choice to address the "Textual Dominance" problem in VLMs and to achieve explicit Cross-Modal Alignment.
>
> We clarify the advantages of this design as follows:
>
> 1. Explicit Vision-Language Alignment (Bridging the Modality Gap)
> If we were to contrast $l_t$ with an augmented version of itself ($l_t^+$), the objective would become a unimodal contrastive learning task (text-to-text). While this might improve the robustness of the textual representation, it fails to explicitly align the LLM's latent space with the visual modality.
> In our framework, the Personalized Multimodal Alignment (PMA) module aims to force the textual features to be semantically close to their corresponding visual features. By treating the visual representation $v$ as the positive anchor for the textual representation $l_t$, we explicitly guide the LLM to encode visual semantics, thereby bridging the gap between the frozen vision encoder and the LLM.
>
> 2. Mitigating Textual Dominance
> As discussed in our Introduction and Section 3.4, a key limitation of existing VLM-based recommenders is that they tend to "underutilize visual features... relying instead on textual descriptions" (lines 51-53). Visual cues often fade away in the deeper layers of the LLM.
> By enforcing $l_t$ to be similar to $v^+$ and $v^h$ in Eq. 7, we introduce a strong regularization signal that prevents the LLM from ignoring the visual input. This ensures that the final recommendation representation preserves critical visual details (e.g., style, color, shape) that are often lost in text-only processing.
>
> 3. Injecting Personalized Visual Preferences ($v^h$)
> A unique advantage of our design is the inclusion of $v^h$ (visual features of user history) as positive samples. By contrasting the target text $l_t$ with the user's historical *visual* interactions ($v^h$), we go beyond simple image-text matching. This forces the LLM's predicted representation to align with the user's specific visual preference patterns learned by the PVRL module. Using $l_t^+$ would simply align the target text with itself, missing the opportunity to reshape the embedding space based on the user's visual history.
>
> In summary, using $v$ instead of $l_t^+$ allows our model to actively correct the modality imbalance in VLMs and directly inject personalized visual preferences into the recommendation logic.

---

> ### Author Response · Authors · 2025-11-22
> **Response to Reviewer zYZi - 3**
>
> > Q3: The "PerVRA" and "PerVLA" are used mixed in multiple places, e.g., Equation 8.
>
> We appreciate the reviewer for pointing this out. We will correct the errors and reflect the changes in the revised version. the correct term throughout the paper should be “PerVRA”.
>
> > Q4: On H&M dataset in Table3, PerVRA's text-only evaluation (HR@5=0.079) is even slightly higher than multimodal evaluation (HR@5=0.078). This contradicts core motivation that "visual information is important”
>
> The importance of visual information for the H&M dataset is clearly evidenced by the baseline performance. As shown in Table 3, the baseline UniMP suffers a drastic performance drop (\~15% drop, from 0.064 to 0.054 in HR@5) when visual modalities are removed. Considering the small magnitude of the gap (\~0.001) in PerVRA, it is evident that PerVRA achieves strong performance by effectively leveraging visual information.
>
> While the H&M dataset is generally visually dependent, some test examples are relatively less influenced by visual information, which can explain small performance differences (~0.001) between text-only and multimodal settings. For instance, in the Figure 5\(c\), the user’s history consists mostly of single-color fashion items, whereas the target item features more vibrant and diverse colors. In the Figure 5\(a\), the history includes flashy clothing, allowing the model to accurately predict the target item.

---

> ### Author Response · Authors · 2025-11-28
> **A Kind Reminder**
>
> Thank you again for your valuable feedback. We have carefully addressed all the comments and revised the manuscript accordingly. We would appreciate it if you could kindly verify that your concerns have been fully resolved in the updated version.
>
> If you have any additional questions or remaining concerns, please feel free to let us know—we are more than happy to clarify further.
>
> Thank you for your time and consideration.

---

### Official Review · Reviewer_SWiu · 2025-11-03

**Soundness:** 2
**Presentation:** 3
**Contribution:** 2
**Rating:** 4
**Confidence:** 5

**Summary:**

This paper introduces PerVRA, a personalized visual representation alignment framework addressing two key issues in VLM-based recommendation: (1) frozen vision encoders lack user-specific preference modeling; (2) VLMs underutilize visual signals and over-rely on text. PerVRA consists of PVRL and PMA modules with dual contrastive learning, treating user history as both positives and hard negatives. The training objective combines task loss and contrastive losses without adding inference cost. Experiments on multiple datasets show strong gains over SOTA baselines and robustness to missing modalities.

**Strengths:**

- Proposes a novel dual-role use of user history (as both positive and negative) for contrastive learning at both visual (PVRL) and textual (PMA) levels, moving beyond standard image-text matching or InfoNCE.
- Outperforms UniMP in sequential recommendation across Amazon/H&M/Netflix/Book, and shows improvements across search, selection, preference prediction, and explanation tasks; robust under missing modality settings.
- Objective functions and pipeline are clearly illustrated, with t-SNE plots supporting the method’s effect on latent structure.
- Demonstrates practical value for making vision "count" in unified VLM frameworks for personalized recommendation, without adding inference cost.

**Weaknesses:**

1. Results are mostly single-run without variance/confidence intervals or t-tests, making it hard to judge reproducibility, especially for low-score datasets. Recommend reporting mean ± std over 3–5 seeds.
2. Hyperparameters like λ₁, λ₂, τ, and MLP size are fixed but not analyzed. The effect of sampling ratio and strategy (e.g., history as hard negatives) is also unclear.
3. Equation (9) introduces a repetition penalty term, but its intuition and difference from standard diversity constraints are insufficiently explained.
4. The mechanism for selecting historical items as negatives is not fully detailed (e.g., sampling ratio, subsampling, temporal decay).
5. Although inference cost is equal, training conditions may differ (e.g., batch size, resolution, cleaning), possibly leading to unfair baseline comparisons.
7. The method is validated on CLIP ViT-L + 3B LLM only. It’s unclear how it generalizes across backbones, model sizes, or alignment layers.

**Questions:**

1. What is the sampling strategy and N for using history items as positives/negatives in UCL/RCL? Is there any N-sweep or heatmap to show trade-offs?
2. How does the repetition penalty differ from typical diversity or de-duplication rules in generation? Any ablation before/after applying it?
3. In missing modality experiments, does randomly dropping half of the images during training reflect real-world distributions? Are text-only test sets matched in candidate composition?
4. PMA aligns visual to the final LLM layer. Has the method tried aligning at cross-attention or intermediate layers (multi-layer distillation)?
5. With 4×A6000 and 10 epochs, how does training cost compare to UniMP? If training cost is equalized, are the gains still significant?
6. Amazon Beauty shows strong gains, but what about visually weak or long-tail domains (e.g., books/comments)? Any cross-domain fine-tuning or zero-shot generalization?

---

> ### Author Response · Authors · 2025-11-22
> **Response to Reviewer SWiu - 1**
>
> > W1: Results are mostly single-run without variance/confidence intervals or t-tests, making it hard to judge reproducibility, especially for low-score datasets. Recommend reporting mean ± std over 3–5 seeds.
>
> To alleviate concerns that our experiments rely solely on a single-run result without reporting variance, we conducted extensive additional experiments using five random seeds. These results show that the proposed PerVRA framework consistently improves both performance and stability over the UniMP baseline.
>
> | Metric | UniMP Mean ± Std | PerVRA Mean ± Std |
> | :--- | :--- | :--- |
> | HR@5 | $0.0452 \pm 0.0167$ | $0.0673 \pm 0.0068$ |
> | NDCG@5 | $0.0249 \pm 0.0092$ | $0.0433 \pm 0.0048$ |
> | MRR@5 | $0.0184 \pm 0.0070$ | $0.0356 \pm 0.0045$ |
>
> > W2: Hyperparameters like λ₁, λ₂, τ, and MLP size are fixed but not analyzed. The effect of sampling ratio and strategy (e.g., history as hard negatives) is also unclear.
>
> We sincerely thank the reviewer for the constructive suggestion. *Please note that we use the same hyperparameters for all experiments.* To address the concern regarding hyperparameter sensitivity, we have conducted extensive experiments analyzing the impact of three key components: (a) the depth of the MLP projection layer, (b) the loss weight coefficients ($\lambda_1, \lambda_2$), and \(c) the temperature parameter ($\tau$). These new results have been added to **Table 9** and discussed in **Appendix D** of the revised manuscript.
> Our findings are summarized as follows:
> 1. Effect of MLP Depth: We evaluated the model with 1, 2, and 3 MLP layers. As shown in the results, the 2-layer configuration achieves the best performance (HR@5: 0.076). A 1-layer MLP tends to underfit the complex user-item relationships, while a 3-layer model introduces excessive nonlinearity that hinders generalization. The 2-layer structure provides the optimal balance for encoding user-specific visual semantics.
> 2. Effect of Loss Weights: We analyzed the trade-off between the user-oriented contrastive loss ($\mathcal{L}\_{\text{UCL}}$, weighted by $\lambda\_1$) and the recommendation-oriented contrastive loss ($\mathcal{L}\_{\text{RCL}}$, weighted by $\lambda_2$). The experimental results indicate that setting $\lambda_1 = 1.5$ and $\lambda_2 = 0.5$ yields the highest accuracy. This suggests that emphasizing user-level preference alignment (UCL) is crucial for building coherent representations, while a moderate weight on RCL ensures fine-grained discrimination without over-differentiation.
> 3. Effect of Temperature ($\tau$): We tested $\tau$ values of $\{0.05, 0.07, 0.1\}$. The model achieves peak performance at **$\tau = 0.07$**. Lower temperatures (e.g., 0.05) lead to instability by over-focusing on hard negatives, whereas higher temperatures (e.g., 0.1) overly smooth the distribution, reducing discriminability.
> These sensitivity analyses confirm that our chosen hyperparameters are robust and effective for the proposed framework.
>
> > W3: Equation (9) introduces a repetition penalty term, but its intuition and difference from standard diversity constraints are insufficiently explained.
>
> We introduce the repetition penalty in Equation (9) to address a specific training dynamic unique to Generative Multimodal Recommendation, distinguishing it from standard inference-time diversity techniques.
> 1. Intuition: Counteracting History Bias from Reconstruction Objectives
> The core intuition stems from our model's use of a History Reconstruction term. While this objective is crucial for ensuring the model accurately encodes user historical preferences, we observed that it introduces a "History Bias." Because the model is trained to reconstruct the history sequence $u_n = \{h_1, ...\}$, it tends to over-optimize for replicating these past items, erroneously predicting them as the target recommendation $y_{>n}$. The repetition penalty in Equation (9), represented as $-\log(1 - p(h|u_{\le n}))$, serves as a structural counter-balance. It explicitly penalizes the model whenever it assigns high probability to items $h$ present in the user's history during the target prediction phase. This forces the model to learn a critical distinction: it must *understand* the history (via reconstruction) but *recommend* novel items (via the penalty), ensuring the generative process focuses on future preferences rather than past repetitions.
> 2. Difference from Standard Diversity Constraints
> Standard methods for repetition avoidance—such as n-gram penalties in Beam Search or Nucleus Sampling—are primarily inference-stage constraints. These techniques act as post-processing filters to mechanically prevent text repetition during decoding.
> In contrast, our proposed repetition penalty is integrated directly into the training objective. This approach prevents "History Bias" at the source, encouraging the model to learn representations that inherently distinguish between "history" and "target" items, rather than simply suppressing valid tokens during inference.

---

> ### Author Response · Authors · 2025-11-22
> **Response to Reviewer SWiu - 2**
>
> > W4: The mechanism for selecting historical items as negatives is not fully detailed (e.g., sampling ratio, subsampling, temporal decay).
>
> Our sampling strategy follows that of UniMP. To capture users’ preference dynamics, we adopt a recency-based sampling strategy rather than random sampling. Specifically, we use the last N interactions from a user’s entire interaction sequence as the history. While the last N historical items from other users always serve as negatives, the last N historical items from the same user play a dual role—acting as positives in UCL and as hard negatives in RCL, as described in Section 3.2. If there are any additional questions or points requiring clarification, we would be glad to address them.
>
> > W5: Although inference cost is equal, training conditions may differ (e.g., batch size, resolution, cleaning), possibly leading to unfair baseline comparisons.
>
> All models share the same training protocol, including batch size, resolution, and data-cleaning procedures. PerVRA differs only by incorporating lightweight 2-layer MLP on the visual features and adding the PVRL and PMA losses defined in Eq. 8. We compare the actual training wall-clock time of our method against the baseline UniMP on the same hardware environment (4x A6000 GPUs) for 10 epochs (approx. 5,690 iterations per epoch).
> • UniMP (Baseline): 1 hour 18 minutes 33 seconds
> • PerVRA (Ours): 1 hour 25 minutes 44 seconds
> The results show that PerVRA incurs only a ~9.0% increase in training time compared to the baseline. Given the significant performance gains observed across diverse tasks, we believe this slight increase in training cost is highly justifiable.
>
> > W6: The method is validated on CLIP ViT-L + 3B LLM only. It’s unclear how it generalizes across backbones, model sizes, or alignment layers.
>
> We respectfully point out that our proposed framework, PerVRA, is designed to be model-agnostic and generalizes well across different model capacities.
>
> 1. Generalization across Model Sizes :
> To address the concern regarding model size dependency, we conducted an additional experiment applying PerVRA to a 1B parameter LLM, which is significantly smaller than the 3B model used in the main paper. As shown in the additional results (attached table), PerVRA achieves remarkable performance gains over the UniMP baseline even on the 1B model:
>
> | Method | HR@5 | NDCG@5 | MRR@5 |
> | :--- | :--- | :--- | :--- |
> | UniMP-1B | 0.0400 | 0.0238 | 0.0183 |
> | **Ours-1B** | **0.0691** | **0.0426** | **0.0340** |
>
>
> 2. Generalization across Backbones:
> PerVRA operates on the representation space after modality encoders via Dual Personalized Contrastive Learning (PVRL & PMA), rather than modifying the internal architecture of the backbones. This modular design allows it to be applied to various vision encoders and LLMs regardless of their specific architectures or alignment layers. To empirically validate this, we applied PerVRA to the Qwen2.5-VL backbone and compared it against the UniMP implementation on the Qwen backbone for the sequential recommendation task. We observed clear performance improvements when PerVRA was applied.
>
> | | HR@5 | NDCG@5 | MRR@5 |
> | :--- | :---: | :---: | :---: |
> | **Qwen2.5-VL + UniMP** | 0.037 | 0.019 | 0.015 |
> | **Qwen2.5-VL + PerVRA** | 0.043 | 0.028 | 0.017 |
>
>
> 3. Alignment Layers in LLMs.
> As shown in Figure 1(a) and consistent with prior findings [B, C, D], visual information gradually diminishes in the later layers of LLMs. Consequently, aligning visual features with the final LLM layer yields the best performance. We conduct an ablation study evaluating alignment at different LLM layers and report the corresponding results below. As expected, we observe that alignment at the last layer shows the best results.
>
> | Align. Layers | HR@5 | NDCG@5 | MRR@5 |
> | :--- | :--- | :--- | :--- |
> | Third-to-Last Layer | 0.0592 | 0.0431 | 0.0379 |
> | Second-to-Last Layer | 0.0625 | 0.0460 | 0.0408 |
> | Last (Final) Layer | 0.0760 | 0.0490 | 0.0410 |
>
> [B] Kaduri, Omri, Shai Bagon, and Tali Dekel. "What's in the Image? A Deep-Dive into the Vision of Vision Language Models." Proceedings of the Computer Vision and Pattern Recognition Conference. 2025.
>
> [C] Yoon, Heeji, et al. "Visual Representation Alignment for Multimodal Large Language Models." arXiv preprint arXiv:2509.07979 (2025).
>
> [D] Fu, Stephanie, et al. "Hidden in plain sight: VLMs overlook their visual representations." arXiv preprint arXiv:2506.08008 (2025).

---

> ### Author Response · Authors · 2025-11-22
> **Response to Reviewer SWiu - 3**
>
> > Q1: What is the sampling strategy and N for using history items as positives/negatives in UCL/RCL? Is there any N-sweep or heatmap to show trade-offs?
>
> We sincerely thank the reviewer for the constructive feedback. Our sampling strategy follows that of UniMP. We address the points raised regarding 1) the sampling strategy for history items and 2) the sensitivity of performance with respect to the number of history items (N) and negatives (B).
>
> **Sampling Strategy for History Items**
>
> To capture users’ preference dynamics, we adopt a recency-based sampling strategy rather than random sampling following UniMP. Specifically, we use the last N interactions from a user’s entire interaction sequence as the history.
>
> **Hyperparameter Sensitivity (N-sweep & Trade-off Analysis)**
> | N, B | HR@5 | NDCG@5 | MRR@5 |
> | :--- | :--- | :--- | :--- |
> | 3, 8 | 0.0574 | 0.0374 | 0.0309 |
> | 5, 5 | 0.0475 | 0.0333 | 0.0287 |
> | 5, 3 | 0.0433 | 0.0288 | 0.024 |
> | 5, 8 | **0.0750** | **0.0451** | **0.0353** |
> | 8, 8 | 0.0516 | 0.0319 | 0.0257 |
>
>
> Following the reviewer’s suggestion, we conducted additional experiments to analyze the performance variations with respect to the number of history items (n) and negative samples (b). The results are summarized below:
>
> 1. **Effect of History Length (N):** With the number of negatives fixed at b=8, increasing the history length from n=3 to n=5 improves HR@5 from 0.0574 to 0.0750, indicating that a moderately longer history helps the model capture user preferences more effectively. However, when the history is further extended to n=8, HR@5 drops to 0.0516, suggesting that an excessively long history may introduce noise or dilute recent user signals, ultimately reducing recommendation performance.
> 2. **Effect of Negative Samples (B):** With history length fixed at n=8, increasing the number of negative samples from b=3 → 5 → 8 led to continuous performance improvement (0.0433 → 0.0475 → 0.0760). This demonstrates that a sufficient number of negatives is crucial for maximizing discriminability in contrastive learning.
>
>
> > Q2: How does the repetition penalty differ from typical diversity or de-duplication rules in generation? Any ablation before/after applying it?
>
> 1. **Difference from Common Diversity Techniques and Its Role as a Training Objective**
>
> In typical generative models, repetition avoidance (de-duplication) or diversity-promoting rules (e.g., n-gram penalties in Beam Search, Nucleus Sampling) are primarily applied at the inference stage, serving as post-processing or decoding constraints to prevent mechanical repetition in generated text.
>
> In contrast, our proposed Repetition Penalty is directly integrated into the training-stage loss function (see Eq. 9), which represents a fundamental difference.
>
> - Motivation: Our model jointly performs the “History Reconstruction” task to accurately learn user preferences. However, this can cause the model to over-optimize for generating past history items, leading to a “History Bias” problem where it simply replicates previously interacted items even when predicting target items.
> - Mechanism: To address this, we explicitly add a penalty term during target item probability calculation for items included in the user history $h_1, ..., h_n$ (Eq. 9). This structurally encourages the model to distinguish between “understanding past history” and “recommending new items” during training. Unlike simple repetition avoidance, this serves as a training constraint specifically designed to prevent recommending items already consumed by the user.
>
> 2. **Effectiveness Demonstrated via Ablation Study (Table 4)**
>
> As the reviewer suggested, we conducted an ablation study in Table 4 to verify the effectiveness of this module. The results indicate that the Repetition Penalty plays a critical role in model performance, particularly for ranking-based metrics.
>
> - Experimental results (on the Amazon Reviews dataset):
>     - Ours (Full Model): HR@5 = 0.076, NDCG@5 = 0.049
>     - w/o Repetition Penalty: HR@5 = 0.068, NDCG@5 = 0.042
> - Analysis: Removing the Repetition Penalty led to a roughly 14.3% drop in NDCG@5. This indicates that without the penalty, the model tends to recommend previously interacted items at top ranks, reducing the recommendation accuracy for new items that the user actually prefers.
>
> Therefore, the proposed Repetition Penalty is essential for enabling generative recommendation models to provide diverse and accurate recommendations without being trapped by historical interactions.

---

> ### Author Response · Authors · 2025-11-22
> **Response to Reviewer SWiu - 4**
>
> > Q3: In missing modality experiments, does randomly dropping half of the images during training reflect real-world distributions? Are text-only test sets matched in candidate composition?
>
> **(1) Does the setting of randomly dropping 50% of images during training reflect real-world data distributions?**
>
> We agree that in real-world service environments, image missingness may not occur randomly at a 50% probability. The reason we adopted the setting of withholding 50% of images during training is not to precisely simulate a particular real-world distribution, but rather to serve as an experimental simulation aimed at maximizing model robustness and validating its stability.
>
> 1. Stress testing for robustness: In practice, images may frequently be missing or unavailable. We wanted to verify whether the model can maintain stable performance even under such incomplete information scenarios. By deliberately blocking image information during training, the model is encouraged to learn aligned latent representations from text even when visual information is absent.
> 2. Enhancing PMA module learning: The 50% ratio is a hyperparameter chosen to balance the learning between ‘cases with visual information’ and ‘cases without visual information,’ thereby helping the proposed PMA (Personalized Multimodal Alignment) module to effectively encode textual data with visual cues.
>
> In conclusion, this setting is not intended to follow the missingness statistics of a specific dataset. Rather, it is a ‘stress test’ designed to improve the model’s adaptability under incomplete visual information and to evaluate its performance under such conditions.
>
> **(2) Is the candidate composition for the text-only test set matched?**
>
> Yes, it is. In our experiments, both text-only evaluation and multimodal evaluation were conducted on the same test set (the same user-item interactions and candidate set). With the same test set, we varies input for evaluation:
>     - Multimodal Evaluation: Inference is performed using all available image information.
>     - Text-only Evaluation: Inference is performed on the same samples using only textual information, excluding images.
>
> Therefore, the candidate composition and ground truth are perfectly aligned between the two evaluation settings. This controlled setup allows a fair comparison of performance changes solely due to the missing modality. Results in Table 3 demonstrate that PerVRA experiences minimal performance degradation even under text-only input.
>
> > Q4: PMA aligns visual to the final LLM layer. Has the method tried aligning at cross-attention or intermediate layers (multi-layer distillation)?
>
>
> As shown in Figure 1(a) and consistent with prior findings [B, C, D], visual information gradually diminishes in the later layers of LLMs. Consequently, aligning visual features with the final LLM layer yields the best performance. We conduct an ablation study evaluating alignment at different LLM layers and report the corresponding results below. As expected, we observe that alignment at the last layer shows the best results.
>
> | Align. Layers | HR@5 | NDCG@5 | MRR@5 |
> | :--- | :--- | :--- | :--- |
> | Third-to-Last Layer | 0.0592 | 0.0431 | 0.0379 |
> | Second-to-Last Layer | 0.0625 | 0.0460 | 0.0408 |
> | Last (Final) Layer | 0.0760 | 0.0490 | 0.0410 |
>
>
> [B] Kaduri, Omri, Shai Bagon, and Tali Dekel. "What's in the Image? A Deep-Dive into the Vision of Vision Language Models." Proceedings of the Computer Vision and Pattern Recognition Conference. 2025.
>
> [C] Yoon, Heeji, et al. "Visual Representation Alignment for Multimodal Large Language Models." arXiv preprint arXiv:2509.07979 (2025).
>
> [D] Fu, Stephanie, et al. "Hidden in plain sight: VLMs overlook their visual representations." arXiv preprint arXiv:2506.08008 (2025).

---

> ### Author Response · Authors · 2025-11-22
> **Response to Reviewer SWiu - 5**
>
> > Q5: With 4×A6000 and 10 epochs, how does training cost compare to UniMP? If training cost is equalized, are the gains still significant?
>
> We thank the reviewer for raising this important point regarding the training efficiency of our model. We understand the concern that introducing multiple contrastive losses and modules might lead to significant computational overhead. However, both our empirical training time  measurements and computational complexity analysis demonstrate that the additional cost is marginal.
>
> 1. Empirical Evidence: Marginal Training Time Increase
> To validate this experimentally, we compared the actual training wall-clock time of our method against the baseline UniMP on the same hardware environment (4x A6000 GPUs) for 10 epochs (approx. 5,690 iterations per epoch).
> • UniMP (Baseline): 1 hour 18 minutes 33 seconds
> • PerVRA (Ours): 1 hour 25 minutes 44 seconds
> The results show that PerVRA incurs only a ~9.0% increase in training time compared to the baseline. Given the significant performance gains observed across diverse tasks, we believe this slight increase in training cost is highly justifiable.
>
>
> 2. Computational Analysis: Efficient Computation via Shared Matrices
> Although PerVRA incorporates four loss terms($\mathcal{L}\_{\text{UCL}}^{v},\mathcal{L}\_{\text{RCL}}^{v},\mathcal{L}\_{\text{UCL}}^{t},\mathcal{L}\_{\text{RCL}}^{t}$), they do not require four independent, expensive contrastive computations. The user-oriented (UCL) and recommendation-oriented (RCL) objectives within the same modality share the same set of embeddings and similarity calculations.
> Specifically, for a batch size $B$, history length $H$, and embedding dimension $D$, the similarity matrix computation for one modality scales as $(BH+B^2)D$. Since PerVRA aligns two modalities (visual and text), the total cost is roughly $2(BH+B^2)D$. By reusing the pairwise similarity matrices for both UCL and RCL objectives, we eliminate redundant computations, keeping the overhead minimal. Furthermore, the additional architecture introduced—a 2-layer MLP for the visual projection 1—is computationally lightweight compared to the backbone Large Language Model. We set $B=8$ and $H=5$.
>
>
> 3. No Additional Inference Cost
> Finally, we wish to emphasize that the proposed Personalized Visual Representation Alignment (PerVRA) modules and contrastive objectives are applied only during the training phase. As stated in the paper, our method introduces no additional cost at inference time3, ensuring that the model remains as efficient as the baseline during actual deployment.
>
> > Q6: Amazon Beauty shows strong gains, but what about visually weak or long-tail domains (e.g., books/comments)? Any cross-domain fine-tuning or zero-shot generalization?
>
> We conducted experiments on the Amazon Books dataset under the same settings. The Books dataset is a challenging domain with a strong long-tail distribution. Nevertheless, the proposed method consistently improved performance across all metrics. This demonstrates that our model can also enhance general recommendation performance in long-tail environments.
>
> | Model | HR@5 | NDCG@5 | MRR@5 |
> | :--- | :--- | :--- | :--- |
> | UniMP | 0.0209 | 0.0172 | 0.0156 |
> | PerVRA | 0.0219 | 0.0179 | 0.0169 |

---

> ### Author Response · Authors · 2025-11-28
> **A Kind Reminder**
>
> Thank you again for your valuable feedback. We have carefully addressed all the comments and revised the manuscript accordingly. We would appreciate it if you could kindly verify that your concerns have been fully resolved in the updated version.
>
> If you have any additional questions or remaining concerns, please feel free to let us know—we are more than happy to clarify further.
>
> Thank you for your time and consideration.

---

### Official Review · Reviewer_189T · 2025-11-04

**Soundness:** 3
**Presentation:** 2
**Contribution:** 2
**Rating:** 4
**Confidence:** 3

**Summary:**

The paper points out that there are two key problems in current VLM recommendation tasks: 1) the features extracted by visual encoders are universal perception-oriented and lack personalization;2) visual features are underestimated in the LLM layer, and recommendation systems rely too much on textual information. To this end, PerVRA has introduced two modules: PVRL (Personalized Visual Representation Learning) and PMA (Personalized Multimodal Alignment), which respectively enhance the personalized expression of visual features through double contrast learning goals and establish more effective alignment between vision and text.

**Strengths:**

+ The motivation of VLM's bias in recommendations is interesting.
+ The evaluation is extensive, and the experimental results look promising.

**Weaknesses:**

+ Existing VLM-based methods often underestimate visual features of user-interacting items in later LLM layers, which is only an empirical inference and lacks rigorous proof.
+ The reason behind the design choice of the method is not clearly explained.
+ There are some presentation errors in the paper, such as line 329.
+ PerVRA and PerVLA alternately appear, which makes it confusing.
+ Lack of comparison with other methods that focus on personalized visual modeling limits the method's innovative evaluation of visual personalization

**Questions:**

+ Why Text-only is better than Text+image is contrary to the conclusion in UniMP.
+ Why can dual contrastive learning objectives balance the contributions of visual and textual features and avoid bias?
+ Multiple contrastive losses and modules are introduced during the training phase, and the actual training costs are not detailed.

---

> ### Author Response · Authors · 2025-11-22
> **Response to Reviewer 189T - 1**
>
> > W1: Existing VLM-based methods often underestimate visual features of user-interacting items in later LLM layers, which is only an empirical inference and lacks rigorous proof.
>
> We thank the reviewer for this insightful comment. Our argument is supported by both prior in-depth analytical studies and extensive quantitative and qualitative experiments conducted in this work. The concrete evidence is as follows.
>
> 1. Consistent Observations in Prior Literature: As discussed in the Introduction, this phenomenon is not an isolated claim made only in our work. Recent studies that deeply investigate the influence of visual information in LLM layers—such as Kaduri et al. (2025) [B], Yoon et al. (2025) [C] and Fu et al. [D], have already reported that visual cues often become diminished relative to textual tokens within VLMs. Our work builds upon this well-documented “textual dominance” phenomenon and shows that the same issue persists in recommendation scenarios.
>
> [B] Kaduri, Omri, Shai Bagon, and Tali Dekel. "What's in the Image? A Deep-Dive into the Vision of Vision Language Models." Proceedings of the Computer Vision and Pattern Recognition Conference. 2025.
>
> [C] Yoon, Heeji, et al. "Visual Representation Alignment for Multimodal Large Language Models." arXiv preprint arXiv:2509.07979 (2025).
>
> [D] Fu, Stephanie, et al. "Hidden in plain sight: VLMs overlook their visual representations." arXiv preprint arXiv:2506.08008 (2025).
>
>
> 2. Text-only vs. Multimodal Inference: Our experiments reveal that the SOTA model UniMP actually performs better—or at least comparably—when using text-only input compared to multimodal (text + image) input. As shown in Fig. 1(a), on the Amazon Reviews dataset, UniMP with text-only inference even outperforms its multimodal counterpart. This performance reversal indicates that visual information is not being effectively utilized by the LLM.
>
> 3. Latent Space Visualization: Our t-SNE visualizations provide intuitive evidence that existing models fail to associate visual features with user preferences. As shown in Figure 1(b) and Figure 4(a), the latent item representations produced by UniMP appear heavily entangled, without clear separation by category or user preference. In contrast, PerVRA—with visual alignment—shows well-formed clusters (Figure 1\(c\), Figure 4(b)), demonstrating that our method captures user-specific visual preferences (e.g., style, color) whereas the frozen vision encoder in conventional VLMs does not.
>
>
> Therefore, our claim that “VLMs do not fully leverage visual features” is not speculative. It is strongly supported by (1) prior analytical studies, (2) our observed performance paradox, and (3) the latent space visualizations presented in this work.

---

> ### Author Response · Authors · 2025-11-22
> **Response to Reviewer 189T - 2**
>
> > W2: The reason behind the design choice of the method is not clearly explained
>
> We sincerely appreciate the reviewer’s constructive and insightful comments. We identified that existing VLM-based recommendation systems suffer from two fundamental limitations: (1) constraints of general-purpose vision encoders for personalized recommendation, and (2) underutilization of visual information within the LLM. To address these issues, we established the following design principles and structured our model accordingly: personalized visual representation learning via dual contrastive objectives (Sec. 3.2–3.3) and personalized multimodal alignment (Sec. 3.4).
>
> **1. Design Intuition of Dual Contrastive Learning: Two Perspectives on User History (Sec. 3.2)**
>
> We view users’ past interactions as playing a *dual role* in recommendation. This motivates PerVRA employs two complementary objectives: UCL (User-oriented Contrastive Learning) and RCL (Recommendation-oriented Contrastive Learning). As shown in the ablation study (Table 4), removing either UCL or RCL leads to performance drops (especially in NDCG@5 and MRR@5), demonstrating that these seemingly opposing learning goals are, in fact, mutually reinforcing.
>
> - **UCL (Coherence Learning):** Historical items serve as positive signals that reflect a user’s preferences (e.g., style, color, category). By bringing them closer to the target item, the model learns user-level coherence in preferences.
> - **RCL (Discriminability Learning):** At the same time, recommendation aims not just to replicate past behaviors but to predict *new* items the user will engage with. Thus, at recommendation time, historical items must act as negative signals relative to the target item. This prevents simple memorization of past choices and enables recommendation-level discriminability.
>
> **2. Design Intuition of PVRL (Personalized Visual Representation Learning) (Sec. 3.3)**
>
> Prior studies typically use frozen vision encoders such as CLIP, which are trained for generic perception tasks. Consequently, they fail to capture personalized notions of similarity—for example, they may group “a knife and a frying pan” together simply because they are both kitchen tools, without considering user-specific preferences.
>
> To remedy this, we introduce the PVRL module to refine the visual feature space with UCL and RCL according to each user’s interaction history. As illustrated in Figure 1\(c\) and Figure 4(b), this leads to meaningful clustering of items based on user preferences.
>
> **3. Design Intuition of PMA (Personalized Multimodal Alignment) (Sec. 3.4)**
>
> Even after improving visual representations, we observed that visual cues tend to be overshadowed by textual signals when passing through LLM layers (textual dominance).
>
> To mitigate this, PMA explicitly aligns the refined visual representations with the LLM’s textual embeddings. This approach overcomes the limitations of prior work—where multimodal models underperform their text-only counterparts (Figure 1(a))—and enables robust performance even when visual inputs are missing.
>
> In summary, each component of PerVRA is not a mechanical addition for performance gains but a necessary design choice grounded in three core goals: defining the dual role of user history, personalizing the visual feature space, and mitigating modality imbalance.
>
> > W3: There are some presentation errors in the paper, such as line 329.
>
> We appreciate the reviewer for pointing this out. We will correct the errors and reflect the changes in the revised version.
>
> > W4: PerVRA and PerVLA alternately appear, which makes it confusing
>
> We appreciate the reviewer for pointing this out. We will correct the errors and reflect the changes in the revised version.

---

> ### Author Response · Authors · 2025-11-22
> **Response to Reviewer 189T - 3**
>
> > W5: Lack of comparison with other methods that focus on personalized visual modeling limits the method's innovative evaluation of visual personalization
>
> We would like to respectfully emphasize that performing *personalized visual modeling* for recommendation within the VLM framework itself is the new paradigm proposed in this work, and to the best of our knowledge, no existing methodology pursues the same objective. Therefore, there is effectively no direct competitor that conducts fine-grained visual personalization *inside* the VLM architecture. We explain the validity of our position and baseline selection through the following three key arguments:
>
> 1. Novelty of Personalized Visual Representation Learning inside VLMs
> As discussed in the introduction, state-of-the-art VLM-based recommendation models such as UniMP and VIP5 rely on a frozen general-purpose vision encoder (e.g., CLIP). While these encoders excel at generic visual *perception* trained on web-scale data, they do not capture *user-specific preferences* such as style, color, or shape. To the best of our knowledge, PerVRA is the first framework to introduce a *Personalized Visual Representation Learning (PVRL)* module for personalized recommendation that refines visual features based on a user’s interaction history.
>
> 2. Comparison with the Best Available Baselines:
> In the absence of direct competitors, we conduct rigorous comparisons with the two most representative groups in adjacent fields, SOTA VLM baselines (UniMP, VIP5+). UniMP and VIP5+ represent the current standard in generative multimodal recommendation. PerVRA improves relative to UniMP, which uses fixed visual features—demonstrating that simply relying on a powerful VLM is insufficient. This evidence shows that our proposed personalized visual modeling is the key driver of performance gains.
>
>
> 3. Establishing a New Perspective on Visual Representations for Personalized Recommendation
> The issue is not a lack of comparison baselines; rather, our work introduces a new conceptual perspective on visual personalization within generative recommendation. As illustrated in Figures 1(b,c) and 4, UniMP’s item embeddings remain entangled, whereas PerVRA produces distinct, preference-aligned clusters that reflect user-specific visual characteristics. This demonstrates that prior methods have not adequately captured personalized visual properties, and PerVRA is the first to explicitly advance this direction.
>
> In summary, the perceived absence of comparisons with “personalized visual modeling” methods arises because PerVRA is the first to overcome the limitations of frozen encoders and introduce this capability inside VLMs. Through extensive comparisons with SOTA VLMs and traditional visual models, we firmly demonstrate both the necessity and effectiveness of this new approach.
>
> > Q1: Why Text-only is better than Text+image is contrary to the conclusion in UniMP.
>
> We appreciate the reviewer’s insightful observation regarding the performance comparison in Figure 1(a). We would like to clarify that this is not due to an experimental error, but rather results from differences in the experimental settings. We believe the reviewer is referring to the numbers of "UniMP w/o Vision" and "UniMP" reported in Table 3 (Ablation Study of UniMP) in their paper. We assume that “UniMP w/o Vision” in their paper refers to a model that never uses visual inputs during either training or inference. In contrast, in our setting of Figure 1(a), we train the model with both vision and text to inspect whether visual information is actually utilized in later LLM layers, but intentionally remove vision only at inference time to evaluate this effect. In summary, the discrepancy stems from differences in the training inputs used in each setting.

---

> ### Author Response · Authors · 2025-11-22
> **Response to Reviewer 189T - 4**
>
> > Q2: Why can dual contrastive learning objectives balance the contributions of visual and textual features and avoid bias?
>
> The dual contrastive learning strategy can balance visual and textual features and prevent bias for the following reasons. Through this strategy, we 1) enhance the visual representations in a user-personalized manner (PVRL), and 2) align the enhanced visual information with textual information (PMA), addressing the bias of VLMs relying excessively on text. The detailed mechanism is as follows:
>
> 1. Enhancing the quality of visual information (PVRL module)
> One of the main reasons existing VLMs are biased toward text is that the features extracted by the visual encoder fail to capture "user preferences" and remain at the level of general perception, making them less useful than text for recommendation tasks. PerVRA first makes visual information more useful through dual contrastive learning (UCL + RCL).
>     - UCL : Treats all of a user’s past interactions as positive, injecting consistent user preferences, such as style or color, into the visual representations.
>     - RCL : Treats past interactions as negatives, enabling the model to distinguish between the target recommendation and historical items.
>     - Through this process, visual features go beyond simple image information and embed user preference information, enabling recommendations that reflect personal preferences.
>
> 2. Direct correction of textual bias (PMA module)
> We notes an "unexpected bias" in VLMs, where, in deeper LLM layers, visual information is underutilized, and the model relies heavily on textual descriptions. To address this, the PMA stage applies dual contrastive learning again:
>     - Multimodal alignment: PMA uses the "personalized visual representations" enhanced in PVRL as guidance to train the text embeddings inside the LLM.
>     - Ensuring balance: By forcing textual features to align with visual features (which now reflect user preferences), the model cannot ignore visual signals when processing text. This encourages text to encode visual cues, balancing contributions from both modalities.
>
>
> In short, dual contrastive learning achieves two goals simultaneously—consistency (UCL) and distinctiveness (RCL)—to improve the quality of visual information. By explicitly aligning it with textual information, it prevents VLMs from relying solely on text and makes visual information a core factor in recommendation decisions.
>
> > Q3: Multiple contrastive losses and modules are introduced during the training phase, and the actual training costs are not detailed.
>
> We thank the reviewer for raising this important point regarding the training efficiency of our model. We understand the concern that introducing multiple contrastive losses and modules might lead to significant computational overhead. However, both our empirical training time  measurements and theoretical complexity analysis demonstrate that the additional cost is marginal.
>
> 1. Empirical Evidence: Marginal Training Time Increase
> To validate this experimentally, we compared the actual training wall-clock time of our method against the baseline UniMP on the same hardware environment (4x A6000 GPUs) for 10 epochs (5,690 iterations).
> • UniMP (Baseline): 78.55 minutes
> • PerVRA (Ours): 85.73 minutes
> The results show that PerVRA incurs only a ~9.0% increase in training time compared to the baseline. Given the significant performance gains observed across diverse tasks, we believe this slight increase in training cost is highly justifiable.
>
>
> 2. Computational Analysis: Efficient Computation via Shared Matrices
> Although PerVRA incorporates four loss terms($\mathcal{L}\_{\text{UCL}}^{v},\mathcal{L}\_{\text{RCL}}^{v},\mathcal{L}\_{\text{UCL}}^{t},\mathcal{L}\_{\text{RCL}}^{t}$), they do not require four independent, expensive contrastive computations. The user-oriented (UCL) and recommendation-oriented (RCL) objectives within the same modality share the same set of embeddings and similarity calculations.
> Specifically, for a batch size $B$, history length $H$, and embedding dimension $D$, the similarity matrix computation for one modality scales as $(BH+B^2)D$. Since PerVRA aligns two modalities (visual and text), the total cost is roughly $2(BH+B^2)D$. By reusing the pairwise similarity matrices for both UCL and RCL objectives, we eliminate redundant computations, keeping the overhead minimal. Furthermore, the additional architecture introduced—a 2-layer MLP for the visual projection layer—is computationally lightweight compared to the backbone Large Language Model. We set $B=8$ and $H=5$.
>
>
> 3. No Additional Inference Cost
> we wish to emphasize that the proposed Personalized Visual Representation Alignment (PerVRA) modules and contrastive objectives are applied only during the training phase. As stated in the paper, our method introduces no additional cost at inference time, ensuring that the model remains as efficient as the baseline during actual deployment.

---

> ### Author Response · Authors · 2025-11-28
> **A Kind Reminder**
>
> Thank you again for your valuable feedback. We have carefully addressed all the comments and revised the manuscript accordingly. We would appreciate it if you could kindly verify that your concerns have been fully resolved in the updated version.
>
> If you have any additional questions or remaining concerns, please feel free to let us know—we are more than happy to clarify further.
>
> Thank you for your time and consideration.

---

### Official Review · Reviewer_xxnQ · 2025-11-04

**Soundness:** 2
**Presentation:** 3
**Contribution:** 2
**Rating:** 4
**Confidence:** 3

**Summary:**

This paper aims to address two limitations in existing VLM-based recommendation tasks: (1) visual features are insufficient for capturing personalized user preferences, and (2) visual features are underutilized by current VLM recommendation models. To tackle these issues, the authors propose PerVRA, which consists of a Personalized Visual Representation Learning (PVRL) module and a Personalized Multimodal Alignment (PMA) module, both built upon contrastive learning. The experiments are conducted using the UniMP model and the Amazon review dataset.

**Strengths:**

1. The task is clearly defined, and the limitations are explicitly described with empirical evidence.
2. The paper is easy to read and understand.
3. PerVRA shows significant improvements over baseline approaches.

**Weaknesses:**

1. The main concern is the generalization of this method. Since PerVRA has been specifically designed for UniMP, it is unclear whether it can be applied to or remain effective for other VLM-based recommendation models. For example, if the vision encoder and text encoder are already highly aligned in a VLM, would PerVRA still provide improvements?
2. Figure 1(a) appears to be inconsistent with Table  3. In Figure 1(a), the text-only setting outperforms the multimodal setting, while in Table 3, the text-only setting performs worse than multimodal.
3. There are no hyperparameter sensitivity experiments. As such, it is unclear how changes to $\lambda_1$ and $\lambda_2$ would affect the results.
4. Several typos exist. For instance, in the OpenReview keywords, “Multimodal RecommeXx” should be corrected. In Section 4.1, Line 329, “Book-Crossing () datasets” appears incomplete.

**Questions:**

If a VLM has a strong text encoder, and the visual encoder that is highly aligned with the text encoder, the problem described by the authors, such as “if a user prefers kitchen-related items, objects like knives and frying pans should be embedded closer together rather than treated as distinct classes”, may not occur, since knives and frying pans would already be close in the semantic space. In such a scenario, would the problem that PerVRA aims to address still exist?

---

> ### Author Response · Authors · 2025-11-22
> **Response to Reviewer xxnQ - 1**
>
> > W1: The main concern is the generalization of this method. Since PerVRA has been specifically designed for UniMP, it is unclear whether it can be applied to or remain effective for other VLM-based recommendation models. For example, if the vision encoder and text encoder are already highly aligned in a VLM, would PerVRA still provide improvements?
>
> The PerVRA framework was implemented and evaluated using UniMP as the backbone. UniMP adopts a Flamingo-based architecture, where the pre-trained vision encoder (CLIP ViT-L) is well aligned with the text encoder via gated cross-attention layers. We observe that this backbone achieves the strongest performance. Additionally, we also present results of our method on other VLMs with Qwen2.5-VL [A].
>
> | | HR@5 | NDCG@5 | MRR@5 |
> | :--- | :---: | :---: | :---: |
> | **Qwen2.5-VL + UniMP** | 0.037 | 0.019 | 0.015 |
> | **Qwen2.5-VL + PerVRA** | 0.043 | 0.028 | 0.017 |
>
> From the table, we can observe the performance comparison between “Qwen2.5-VL + UniMP” and “Qwen2.5-VL + PerVRA.” Despite the fact that Qwen2.5-VL may have different pre-trained alignment characteristics compared to Flamingo, applying PerVRA yields clear performance improvements.
>
> [A] Bai, Shuai, et al. "Qwen2. 5-vl technical report." arXiv preprint arXiv:2502.13923 (2025).
>
> > W2: Figure 1(a) appears to be inconsistent with Table  3. In Figure 1(a), the text-only setting outperforms the multimodal setting, while in Table 3, the text-only setting performs worse than multimodal.
>
> We appreciate the reviewer's keen observation. We clarify that the results in Figure 1(a) and Table 3 are not contradictory but rather highlight different aspects of model behavior under distinct experimental setups: Assessing the Contribution of Visual Information **under Standard Training** vs. Robustness Simulation **under Missing-Modality Conditions**.
>
> 1. Figure 1(a): Assessing the Contribution of Visual Information under Standard Training
> - Setup: The model is trained in a standard multimodal setting (using all image-text pairs) and evaluated by toggling visual inputs on/off.
> - Objective: By comparing performance with visual inputs enabled versus disabled, we evaluate the contribution of visual information.
> - Results: This confirms the motivation of our study. In standard VLM baselines (like UniMP), visual representations are not personalized or well-aligned. Consequently, providing visual tokens during inference may not provide meaning information for recommendation, resulting in worse performance compared to using text alone.
>
> 2. Table 3: Robustness Simulation under Missing-Modality Conditions
> - Setup: Unlike Figure 1(a), this experiment adopts a "Missing Modality" training strategy. We randomly mask 50% of the images in the user history during training.
> - Objective: This is not intended to mimic a specific real-world distribution but serves as a stress test simulation to maximize and verify model robustness. By intentionally blocking visual information, we force the model to learn latent representations where textual features are aligned with visual cues, ensuring stability even when images are absent.
> - Results: This result validates the effectiveness of our Personalized Multimodal Alignment (PMA) module. The 50% masking acted as a balanced hyperparameter that trained the PMA module to effectively encode visual cues into the textual representations. Consequently, our model successfully retains visual semantics within the text embeddings, demonstrating superior robustness where performance is preserved even when explicit visual data is removed.
>
> Conclusion:
> Therefore, the results are consistent with our claims:
> - Figure 1(a) shows that simply adding visual features without alignment may not yield meaningful improvements.
> - Table 3 proves that our method (PerVRA) effectively solves this by aligning modalities, allowing the model to be robust ("Text-only" performance is maintained) where baselines fail (UniMP performance drops), even under missing-modality conditions.

---

> ### Author Response · Authors · 2025-11-22
> **Response to Reviewer xxnQ - 2**
>
> > W3: There are no hyperparameter sensitivity experiments. As such, it is unclear how changes to and would affect the results.
>
> We sincerely thank the reviewer for the constructive suggestion. To address the concern regarding hyperparameter sensitivity, we have conducted extensive experiments analyzing the impact of three key components: (a) the depth of the MLP projection layer, (b) the loss weight coefficients ($\lambda\_1, \lambda\_2$), and \(c\) the temperature parameter ($\tau$). These new results have been added to Table 9 and discussed in Appendix D of the revised manuscript.
> Our findings are summarized as follows:
> 1. Effect of MLP Depth: We evaluated the model with 1, 2, and 3 MLP layers. As shown in the results, the 2-layer configuration achieves the best performance (HR@5: 0.076). A 1-layer MLP tends to underfit the complex user-item relationships, while a 3-layer model introduces excessive nonlinearity that hinders generalization. The 2-layer structure provides the optimal balance for encoding user-specific visual semantics.
> 2. Effect of Loss Weights: We analyzed the trade-off between the user-oriented contrastive loss ($\mathcal{L}\_{\text{UCL}}$, weighted by $\lambda\_1$) and the recommendation-oriented contrastive loss ($\mathcal{L}\_{\text{RCL}}$, weighted by $\lambda\_2$). The experimental results indicate that setting $\lambda\_1 = 1.5$ and $\lambda\_2 = 0.5$ yields the highest accuracy. This suggests that emphasizing user-level preference alignment (UCL) is crucial for building coherent representations, while a moderate weight on RCL ensures fine-grained discrimination without over-differentiation.
> 3. Effect of Temperature ($\tau$): We tested $\tau$ values of $\{0.05, 0.07, 0.1\}$. The model achieves peak performance at $\tau = 0.07$. Lower temperatures (e.g., 0.05) lead to instability by over-focusing on hard negatives, whereas higher temperatures (e.g., 0.1) overly smooth the distribution, reducing discriminability.
> These sensitivity analyses confirm that our chosen hyperparameters are robust and effective for the proposed framework.
>
> > W4 Several typos exist
>
> We appreciate the reviewer for pointing this out. We will correct the errors and reflect the changes in the revised version.

---

> ### Author Response · Authors · 2025-11-22
> **Response to Reviewer xxnQ - 3**
>
> > Q1: If a VLM has a strong text encoder, and the visual encoder that is highly aligned with the text encoder, the problem described by the authors, such as “if a user prefers kitchen-related items, objects like knives and frying pans should be embedded closer together rather than treated as distinct classes”, may not occur, since knives and frying pans would already be close in the semantic space. In such a scenario, would the problem that PerVRA aims to address still exist?
>
> The limitations that PerVRA seeks to address are likely to remain even when using a powerful VLM with highly aligned visual and textual encoders. To examine this, we conducted a preliminary study following the reviewer’s example: we test visual encoders of different capacities and measured the embedding similarity among images of knife, hurting knife, and frying pan. As shown below, stronger models (e.g., ViT-G-14) actually amplify the separation, treating these objects as even more distinct and unrelated classes. This indicates that, even with a strong visual encoder well aligned with the text encoder, the underlying limitation persists.
>
> | Model Name | Size | Knife ↔ Hurting Knife<br>*(Semantic Similarity)* | Knife ↔ Frying Pan<br>*(Preference Similarity)* |
> | :--- | :---: | :---: | :---: |
> | **ViT-L-14** | Large | 0.8931 | 0.7720 |
> | **ViT-H-14** | Huge | 0.6321 | 0.5598 |
> | **ViT-G-14** | Giant | 0.5865 | 0.5625 |
>
> In addition, PerVRA enables **user-oriented alignment**, reorganizing item embeddings according to each user’s preferences (e.g., style, color, aesthetic tendencies)—even when a general-purpose visual encoder would arrange them differently. In other words, PerVRA does not only simply cluster semantically similar objects but also re-adjusts the visual representations based on a user’s actual interaction history.

---

> > ### Comment · Reviewer_xxnQ · 2025-11-27
> >
> > Thanks to the authors for the detailed rebuttal, which addresses most of my concerns. I still believe that a sufficiently powerful VLM with highly aligned visual and textual encoder might mitigate this problem. However, I know this point may be beyond the scope of the current paper. Overall, the clarification is helpful, and I am willing to raise my score to 6.

---

> > > ### Author Response · Authors · 2025-11-28
> > > **Thank you for your positive feedback**
> > >
> > > We sincerely appreciate that you revised the score upward after reviewing our rebuttal. Thank you for taking the time to reevaluate our work and for your considerate assessment.
> > >
> > > If you have any additional questions or remaining concerns, please feel free to let us know—we are more than happy to clarify further.

---

### Author Response · Authors · 2025-11-22
**General Response**

We sincerely thank the reviewers for their constructive feedback, which has substantially improved this manuscript. The key changes made based on these comments are highlighted in red within the revised manuscript:

**Robustness Analysis (Reviewer [SWiu]):** We conducted extensive additional experiments using five random seeds in Appendix F to verify the stability of our results. The analysis confirms that our proposed framework achieves statistically significant improvements independent of specific seed selection.

**Robustness across Model Sizes (Reviewer [SWiu]):** We addressed concerns regarding model size dependency by validating our method on a 1B parameter model in Section 4.8. The consistent relative improvements demonstrate that the effectiveness is driven by our personalized alignment mechanism rather than model scale.

**Computation Analysis (Reviewer [189T, SWiu]):** PerVRA adds only minimal training overhead because its additional contrastive objectives reuse similarity matrices and its lightweight projection modules are used only during training. Empirically, training time increases by just ~9%, which is negligible compared to the substantial performance improvements. (Appendix G)


**Clarification on Novelty of Personalized Dual Contrastive Learning (Reviewers [189T, zYZi]):** Our Personalized Dual Contrastive Learning introduces a novel mechanism that simultaneously treats user history as both positive (for preference coherence) and negative (for recommendation discriminability). This dual formulation enables personalized visual representation learning and balanced multimodal alignment within the VLM.

**Sensitivity Analysis (Reviewers [xxnQ, SWiu, zYZi]):** We added a comprehensive sensitivity analysis in Appendix D regarding the number of MLP layers, loss weight parameters, and temperature. The results empirically justify our design choices for optimal user preference alignment and recommendation discriminability.

**Generalizability Analysis (Reviewer [xxnQ]):** We extended our experiments to the Qwen architecture in Section 4.7 to demonstrate the generalizability of our framework. The consistent performance gains verify that our approach is effective across diverse VLM backbones, not just Flamingo-based models.


**Typo Corrections:** Throughout the manuscript, we have carefully corrected typos and grammatical errors to enhance readability.

In the following sections, we present detailed responses to the specific points raised by each reviewer. We are grateful for the insightful suggestions that have helped refine our work, and we trust that the revisions and additional experiments effectively resolve the concerns. We remain open to further discussion regarding any additional questions

---

### Author Response · Authors · 2025-12-01
**Official Comment by Authors**

## Dear Area Chair,

We would like to briefly summarize the discussion that took place during the rebuttal period.

Reviewers initially raised concerns regarding statistical robustness across random seeds and model sizes, the computational overhead of training, the novelty of our personalized dual contrastive learning, hyperparameter sensitivity, generalizability across different VLM backbones, and presentation errors (typos), which we addressed thoroughly in our response. Reviewer xxnQ indicated that these concerns were fully resolved and subsequently increased their score to 6.

Although we were unable to receive follow-up responses from reviewers 189T, SWiu, and zYZi, their concerns on generalization, hyperparameter sensitivity, and presentation quality were aligned with those of Reviewer xxnQ, and were addressed in the same discussion.

We also carefully responded to all remaining comments and incorporated the corresponding improvements, including comprehensive typo corrections, into the revised manuscript.

Thank you very much for your thoughtful review and effort.

---

### Meta-Review · Area_Chair_ZLzf · 2026-01-09

**Summary:**

This paper initially received negative ratings (4) from all reviewers. The author rebuttal addressed many concerns, including clarification of methodological details, insights into the methodological design, experimental details, additional ablation studies, and statistical tests, and one reviewer raised their rating to 6 after considering author rebuttal. These responses have improved the paper and made it a borderline submission, resulting in a challenging decision.

In my opinion, the overall methodological novelty, insights, writing quality, and clarity of this paper remain slightly below the ICLR acceptance threshold relative to other submissions, and I therefore recommend rejection. Nonetheless, I am open to my recommendation being reconsidered if the broader committee feels otherwise.

**Reviewer Concerns:**

It appears to me that most of the reviewers' concerns have been addressed.

**Reviewer Scores:**

Reviewer xxnQ explicitly stated that they will raise their score from 4 to 6 after considering the author rebuttal.

It is difficult to predict how the other three reviewers will adjust their scores (their initial ratings are all 4), as their responses to the author rebuttal may vary.

Overall, I think this paper is likely to receive a mix of positive and negative reviews.

---

### Decision · Program_Chairs · 2026-01-26

Reject